# Uncertainties in climate change projections covered by the ISIMIP and CORDEX model subsets from CMIP5

Ito, Rui[1,2], Hideo Shiogama[3], Tosiyuki Nakaegawa[2], Izuru Takayabu[2]

[1]Japan Meteorological Business Support Center, Tsukuba, 305-0052, Japan
[2]Meteorological Research Institute, Japan Meteorological Agency, Tsukuba, 305-0052, Japan
[3]National Institute for Environmental Studies, Tsukuba, 305-0053, Japan

*Correspondence to*: Rui Ito (rui.ito@jmbsc.or.jp)

**Abstract.** Two international projects, ISIMIP (Inter-sectoral Impact Model Inter-comparison Project) and CORDEX (Coordinated Regional Climate Downscaling Experiment), have been established to assess the impacts of global climate
change and improve our understanding of regional climate, respectively. Model selection from the GCMs (general circulation models) within CMIP5 (fifth phase of the Coupled Model Inter-comparison Project) was conducted by the different approaches for each project: one is a globally consistent model subset used in ISIMIP and another is a region-specific model subset for each region of interest used in CORDEX. We evaluated the ability to reproduce the regional climatological state by comparing the subsets with the full set of CMIP5 multimodel ensemble. We also investigated how well the subsets captured the uncertainty
in the climate change projected by the full set, to provide increased credibility for the scientific outcomes from each project. The spreads of the biases and Taylor's skill scores from the ISIMIP and CORDEX subsets are smaller than that from the full set for the regional means of surface air temperature and precipitation. However, the ISIMIP and CORDEX subsets show the larger spread than high-performance models from the full set, despite using a small number of models in ISIMIP and CORDEX. It was shown that better subsets exist that would have smaller biases and/or higher scores than the current subset. The ISIMIP
subset captures the uncertainty range of the regional mean of temperature change projections by the full set better than the CORDEX subsets in 10 of 14 terrestrial regions worldwide. Compared with 10,000 randomly selected subset samples, the CORDEX subset shows low coverage of the uncertainty for the temperature change projections in some regions, and the ISIMIP subset high coverage in all regions. On the other hand, for the precipitation change projections, the CORDEX subsets show lower coverage in half of the regions than the randomly selected subsets, but tend to cover the uncertainty wider than the
ISIMIP subset. In the regions where CORDEX used nine models or more, good coverage (>50%) is evident for the projections of both temperature and precipitation. The globally consistent model subset used in ISIMIP could have difficulty in capturing uncertainties in the regional precipitation change projections, whereas it widely covers uncertainties in the temperature change projections. The region-specific model subset, like CORDEX, can cover the uncertainties in both temperature and precipitation changes well compared to the global common subset, but a large number of models is needed. By changing the number of
models from the current ensemble members to at least nine members, high coverage for both uncertainties can be also obtained in the other regions and this information would help model selections in the next generations.

## 1 Introduction

A global dataset of climate change projections has been generated by the Coupled Model Inter-comparison Projects (CMIP). Using this dataset, numerous climatological studies have been in progress to advance our understanding of the increasingly severe problems associated with climate change. Regarding regional climate change, dynamical and statistical downscaling experiments have been conducted to create high-resolution climate products derived from the global CMIP dataset via a regional climate model. In addition, impact studies and examinations of adaptation planning have progressed in close parallel with the climate studies, using those climate products at both global and regional scales.

When we conduct an impact assessment of climate change and consider possible adaptation or mitigation measures, the information regarding the largest potential change in the climate is required to consider the most severe states of climate change, in addition to information regarding how the climate changes on average. Although the CMIP multiple global climate model (GCM) ensemble is the ensemble of opportunity and do not necessarily represent the full uncertainty in the climate projections (Knutti, 2010), they are useful for investigating the uncertainty in the future projections. By using the climate projections from the CMIP ensemble, it is at least possible to examine the maximum−minimum climate change scenarios within the ensemble. It is desirable to use GCMs as much as possible to address the most severe problems but, due to limitations in computing resources, relatively small subsets of the models are generally used in regional downscaling studies and impact assessments. The present subset tends to be selected under the conditions that the simulation accuracy is better for the climatological state of interest or the data required for the study is readily available. Methods of specifying the best subset, based on the accuracy of the historical climate simulations and/or capturing the possible maximum range in the variation of projections among the models (hereafter uncertainty), have been proposed (Reichler and Kim, 2008; Cannon, 2015; Mendlik and Gobiet, 2016). The optimum method, however, remains to be determined because the interests depend on the studies, for instance, how the model performance is considered, which climatological or extreme variables are used and which region is interested. When the sample size of a subset is limited, appropriate strategies are necessary to select subsets of GCMs that have smaller biases in the historical climate simulations and cover the widest possible uncertainty range of future projections. Without such a strategy, we might erroneously interpret the information regarding climate change and impact assessment obtained from the subsets.

The inter-sectoral impact model inter-comparison project (ISIMIP; https://www.isimip.org) was designed as a framework to assess the impacts of climate change in different sectors and at different scales (Schellnhuber et al., 2014). This project used consistent climate and socio-economic input data to multiple impact models. Five GCMs were selected in the fast track of ISIMIP: HadGEM2-ES, GFDL-ESM2, IPSL-CM5A-LR, MIROC-ESM-CHEM, and NorESM1-M. The main selection condition was that the climate data generated by the models was available at the relevant stage of the project, with the attempt of broadly capturing the global change in surface air temperature (hereafter referred to as 'temperature' for simplicity) and precipitation (Warszawski et al., 2014; ISIMIP protocol 2018). After that, the five GCMs had been changed to four GCMs in the next round simulations (ISIMIP2b; Frieler et al., 2017) because of a lack of wind data for NorESM1-M and a higher horizontal resolution and the better representation of various fields (e.g., El Niño–Southern Oscillation and the monsoon) in

MIROC5 than in MIROC-ESM-CHEM. A feature of the uncertainty range identified from the five GCMs in the fast track was investigated in detail by McSweeney and Jones (2016) (hereafter MJ2016), who indicated that the subset covers more of the uncertainty in the temperature and precipitation changes projected by 36 CMIP5 GCMs, than other randomly sampled five-GCM subsets. They also illuminated that region-specific subsets generally cover more the uncertainty than globally consistent subsets in 26 global regions.

One subset of GCMs was globally used in ISIMIP, but in the coordinated regional climate downscaling experiment (CORDEX; http://www.cordex.org) project, a GCM subset was selected for each defined region to generate a regional climate dataset for climate studies and impact assessments (Giorgi et al., 2009; Giorgi and Gutowski, 2015). Fourteen regions of interest were defined and subsets of between 3 and 15 GCMs were used for each region. The conditions required here were that input data to a regional climate model (RCM) were available and easily acquired, and they also tended to select GCMs that were developed at the institute located in the region of interest. The advantage of CORDEX is that it enables a regional climate assessment using a dataset from 'optimal' multi-GCMs and multi-RCMs for the region of interest. However, Gutowski et al. (2016) pointed out as one of the problems in the first phase of CORDEX that the different models, especially the number of models, among the regions make difficulty to provide the consistent climate scenario among their regions. Therefore, in the next generation of CORDEX to be included in the sixth phase of CMIP, they have an intention to downscale projections from a core set of GCMs as a minimum model set that is common across the regions, similar to the approach in ISIMIP (CORDEX CORE; Gutowski et al., 2016).

A globally consistent GCM subset will facilitate discussion of climate change and its impacts beyond regional divisions. However, it is unclear whether the globally consistent subset adequately represents the phenomena that characterize the climate in the region of interest. In particular, the spatial pattern of a projected change in precipitation is strongly dependent on the GCMs selected (Giorgi and Gutowski, 2015; McSweeney et al., 2015). Therefore, the possibility of insufficiently capturing the regional climate change and its valid uncertainty could be increased, as noted by MJ2016. In contrast, a region-specific GCM subset can include GCMs which more precisely reproduce the target regional climate (McSweeney et al., 2015). However, it does not enable discussions about the difference among regions and the interaction of impacts across the regions. Although there are advantages to both approaches to select a subset, it is necessary that we understand the characteristics of the current subsets selected using the approaches of the ongoing projects if we are to improve the process in the next generations of the projects.

In this study, we assessed the current subsets of CMIP5 multi-GCM ensemble being used in ISIMIP and CORDEX by clarifying the climatological characteristics expressed by each subset from two points of view: how high the ability to reproduce the historical climate is (i.e., model performance) and to what extent the uncertainty in the projections obtained from the subsets covers the uncertainty from the full set. We examined temperature and precipitation climatologies in a simple method, but the clarification of characteristics is important for understanding the basic nature of dataset and increasing the credibility of the scientific outcomes from each project. In addition, with reference to MJ2016, we also explored whether the

subset used was able to capture the uncertainty from the full set more widely than the other model subsets when using the same sample size.

Regarding the ISIMIP subset, there are two updated points from MJ2016. One is the investigations of the ability to represent historical climate for the ISIMIP subset, which MJ2016 did not mentioned; another is that our target GCMs are four GCMs selected in ISIMIP2b (unless specified otherwise, hereafter refers to as ISIMIP). Regarding the CORDEX subset, previous studies have assessed the GCM simulations in some regions, but are limited (e.g., Haensler et al., 2013 for Africa; Bartók et al., 2017 for Europe; Karmalkar, 2018 for North America). Therefore, even a simple assessment of GCM simulations is needed for understanding their downscaled simulations.

Uniform assessment across regions permits to discuss the regional characteristics and the possibility of heterogeneous scenario among regions as mentioned above. Furthermore, by using both subsets from ISIMIP and CORDEX, we can explore the difference between the original subset in CORDEX and the subset assuming CORDEX CORE (global common subset), which could be helpful information for the model selection in CORDEX CORE.

## 2 Data and Methods

### 2.1 Dataset

We analysed the historical runs of 50 atmosphere–ocean GCMs (AOGCMs) and the Representative Concentration Pathways (RCP) 8.5 scenario runs of 42 AOGCMs participating in CMIP5 (Taylor et al., 2012). A single ensemble member, r1i1p1, was selected for each model, except for CESM1-WACCM (r2i1p1), CSIRO-Mk3L-1-2 (r1i2p1) and EC-EARTH (r8i1p1). It is because the member, r1i1p1, of CESM1-WACCM and CSIRO-Mk3L-1-2 were not available and temperature change from r1i1p1 of EC-EARTH was over two-standard deviation of the changes from the 42 models in more than 60% of our target regions. In the followings, the full set of the multi-GCM ensemble indicates the 50 historical runs when we assessed the ability to reproduce the historical climate (CMIP$_{Full\_Hist}$), while does the 42 future projections which are estimated from both historical and rcp85 runs when we discussed the future projections (CMIP$_{Full\_Future}$).

We compared the simulations of the subsets of GCMs used in ISIMIP and CORDEX with the full ensemble. ISIMIP used four GCMs for their various impact assessments: GFDL-ESM2M, HadGEM2-ES, IPSL-CM5A-LR and MIROC5 (Frieler et al., 2017). On the other hand, CORDEX used the subset in which the combination of GCMs were altered for each defined region. The number of GCMs used in each of the defined regions is listed in Table 1, and each GCM is listed in Supplement 1. The regional classification used to investigate the regional performance and the projection was based on the classification in CORDEX shown in Supplement 2. In this study, we focused on global land area, considering the importance for both programs because of the relevance to human activities.

The analysis periods were the year 1986–2005 (but 1985–2004 for HadGEM2-CC and HadGEM2-ES) for the historical runs and the year 2081–2100 (but 2080–2099 for MRI-AGCM60 and CESM1-WACCM) for the RCP8.5 runs. Monthly mean temperature and precipitation data over these periods were interpolated onto a 2.5° × 2.5° grid for each model. The grid cells

with the temporal mean precipitation of < 0.1 mm/day were defined as 'too dry' grid cells and the precipitation value at the cells was out of consideration. It is because we expressed the precipitation change in a ratio and thus the ratio tends to be large at too dry cells even when the change is quantitatively extreme small. Such a large ratio is difficult to explain its meanings physically. By applying the threshold, the grid cell indicating an extremely large ratio, for instance, 100% were excluded. The

total number of the excluded grid cells is approximately 5% of all target cells as an average over the used members.

To validate the model representations, we compared the simulated estimates with the observed datasets. With respect to precipitation, Sun et al. (2018) highlighted differences among the observational datasets. Consequently, to avoid a misreading of the model performance due to such discrepancies, we used seven different precipitation products that covered the global land area over the period of interest. The observation products were the Climatic Research Unit Timeseries (CRU) v.4.01

(Harris et al., 2014) for temperature and precipitation, and the following for precipitation only: the global unified gauge-based analysis by NOAA Climate Prediction Center (CPC) v.1.0 (Xie et al., 2010), the Global Precipitation Climatology Centre (GPCC) full data reanalysis v.7.0 (Schneider et al., 2016), NOAA's Precipitation reconstruction over Land (PRECL) v.1.0 (Chen et al., 2002), the CPC Merged Analysis of Precipitation (CMAP; Xie and Arkin, 1997), the Global Precipitation Climatology Project (GPCP) v.2.2 (Huffman et al., 2015), and the Multi-Source Weighted-Ensemble Precipitation (MSWEP)

v2.1 (Beck et al., 2019). The difference among the observations was calculated as the deviation from GPCC as the reference. To quantify the ability to reproduce spatial patterns of the observations, we used the skill score proposed by Taylor (2001) (hereafter referred to as skill score) as follows:

$$S = 4(1+R)/\{(\sigma+\sigma^{-1})^2(1+R_0)\}, \tag{1}$$

where $R$ is the spatial correlation coefficient between reference observation and simulation, $\sigma$ is the standard deviation of

simulation normalized by the reference spatial pattern and $R_0$ is the maximum correlation attainable. The value of $R_0$ was assumed to 1 here. In addition to the skill score, we also evaluate the magnitude of the model bias. Using both metrics enables the assessment of both the spatial pattern and the bias magnitude.

## 2.2 Coverage of uncertainty and random selection

Coverage was estimated from a comparison between the full uncertainty range of the projections made by two model sets,

which was defined by McSweeney et al. (2015) as a fractional range coverage, FRC. In this study, we computed the regionally averaged projections for each model, and then the FRC were estimated using the regional averages. The FRC from the regional averages (FRA) was defined as the fraction of the maximum−minimum range of the uncertainty in the regional averaged projections from a subset of CMIP$_{Full\_Future}$ ($R_{Sub}$) to the range from CMIP$_{Full\_Future}$ ($R_{Full}$), as follows:

$$FRA = \frac{R_{Sub}}{R_{Full}}. \tag{2}$$

The range of $R_{Sub}$ was computed from the ISIMIP and CORDEX subsets and also from arbitrary subset samples we generated. From the comparison with the arbitrary samples, we can investigate how well the ISIMIP and CORDEX subsets captured the

uncertainty range of projections. With reference to MJ2016, our arbitrary samples were generated by randomly selected $n$ models without repetition from $CMIP_{Full\_Future}$ 10,000 times, where $n$ is the sample size of subsets in ISIMIP ($n = 4$) or CORDEX ($n$ depends on the regions; see Table 1). Then, the variance of the FRA was estimated from the 10,000 random subset samples of $CMIP_{Full\_Future}$ and compared with the FRA from the ISIMIP and CORDEX subsets.

**3 Results**

**3.1 Performance in reproducing historical climate**

Using model biases and skill scores, we evaluated the historical climate reproduced by the GCM subsets used in ISIMIP and CORDEX. The GCM subsets used in ISIMIP and CORDEX are hereafter referred to as the ISIMIP subsets and CORDEX subsets, respectively. For the evaluations, we also used two high performance subsets: one is composed of models with lower

bias than the 50th percentile (median) of the $CMIP_{Full\_Hist}$ biases; the other is models with higher skill score than the median of the $CMIP_{Full\_Hist}$ scores (referred to $CMIP_{lowB}$ and $CMIP_{highS}$, respectively). The models included in the high performance subset is shown in Supplement 3. $B(v(E))$ and $S(v(E))$ indicate the regional mean biases and skill scores for variable $v$ and ensemble subset $E$, respectively.

Figure 1 shows the model bias associated with the annual mean precipitation in the 14 CORDEX regions over a 20-year period.

Compared with the maximum values of $B(P(CMIP_{Full\_Hist}))$ for the precipitation ($v$=P), the maximum values of $B(P(ISIMIP))$ and $B(P(CORDEX))$ are clearly small, especially in the Mediterranean (MED), Southeast Asia (SEA), and the polar regions. The spreads of $B(P(ISIMIP))$ and $B(P(CORDEX))$ in MED are within the spread of the discrepancy among the observations, which suggests that the model selection works effectively to select models with high ability to reproduce the observed regional mean precipitation quantitatively. However, compared with the high performance subsets, some models in the ISIMIP and

CORDEX subsets have a bias exceeding the maximum values of $B(P(CMIP_{lowB}))$, or $B(P(CMIP_{highS}))$ in some regions, despite the small number of models used in ISIMIP and CORDEX. Therefore, our results indicate that less bias models could be selected than those currently being used. The difference in the spread between the ISIMIP and CORDEX subsets has a characteristic in region-by-region and part of them relates to the overlapping of model members used across ISIMIP and CORDEX. For example, in five regions of Central and South America, Europe, Africa and South Asia, the CORDEX subsets

include more than three of four ISIMIP models and the ensemble is large (Supplement 1). As a result, the variance of biases estimated from the CORDEX subset covers that from the ISIMIP subset. Especially in Europe, the difference of the variance between the CORDEX and ISIMIP subsets is large and it is found that the models used in the CORDEX subset but not included in the ISIMIP subset make the variance increase. Focusing on the regions where the CORDEX subsets include only two models in the ISIMIP subset, the variance from the CORDEX subset tends to be larger than that from the ISIMIP subset, especially in

the regions with large ensemble of the CORDEX subsets, like North America, SEA and Australasia. By contrast, the variance from the CORDEX subsets is relatively small in the regions with small ensemble of the CORDEX subsets, like MENA and

Central Asia. In East Asia, the variance is small in CORDEX despite using seven models in contrast to four models in ISIMIP, indicating that the biases from the seven models are almost the same.

With respect to the spatial pattern of the annual mean precipitation, ISIMIP and CORDEX incorporate some models with a worse score than the minimum value of $S(P(CMIP_{highS}))$ (Supplement 4). That is to say, ISIMIP and CORDEX subsets include
members showing a spatial pattern of low similarity to that of observations. $S(P(ISIMIP))$ and $S(P(CORDEX))$ fall within the observational spread only in the Arctic.

We also assessed model performance for the annual mean temperature ($v$=T). The maximum values of $B(T(ISIMIP))$ and $B(T(CORDEX))$ are smaller, or equal to the maximum value of $B(T(CMIP_{highS}))$ (except for the CORDEX subsets in East Asia and North America), but are larger than the maximum value of $B(T(CMIP_{lowB}))$ (Supplement 5). The spread of
$B(T(ISIMIP))$ is covered by that of $B(T(CORDEX))$ in the same four regions as the bias in the precipitation except for Europe, because of the overlapping of model members used. The spreads of $B(T(ISIMIP))$ and $B(T(CORDEX))$, however, resemble each other compared with the precipitation bias in most regions, indicating that CORDEX used models with a quantitatively similar performance to ISIMIP, despite using more models than ISIMIP except for Central Asia. Both subsets included models with a worse score than the minimum value of $S(T(CMIP_{highS}))$ in 85% of the regions (Supplement 6). Therefore, relative to
$CMIP_{highS}$, the subsets can quantitatively represent the observed temperature as a regional average well but the spatial pattern represented by some members in the subsets does not resemble that of the observations.

Even though the model selections conducted in ISIMIP and CORDEX narrow the spreads of model bias and the score from $CMIP_{Full\_Hist}$, the largest bias and the worst score from the ISIMIP and CORDEX subsets distribute beyond the biases and the scores from high performance models in the full set.

**3.2 Uncertainty range of the projected changes in annual mean temperature and precipitation**

Future projections obtained from the ISIMIP and CORDEX subsets were compared with those from the full set, and also from high performance models, as with the evaluations in Section 3.1. Because the small biases or high skill scores models used in this section are composed of the models included in $CMIP_{Full\_Future}$, we refer as $CMIP'_{lowB}$ and $CMIP'_{highS}$ instead of $CMIP_{lowB}$ and $CMIP_{highS}$. Projected change in annual mean temperature and precipitation are designated by $\Delta T(E)$ and $\Delta P(E)$, respectively.
Figure 2 shows the uncertainty range of the projected temperature increments, calculated from the average over the 20-year period for each model. Although ISIMIP used fewer models than CORDEX, the uncertainty range of $\Delta T(ISIMIP)$ is greater than or equal to that of $\Delta T(CORDEX)$ except for South Asia, Australasia, South America, and Central America. The uncertainty ranges of $\Delta T(CMIP'_{lowB})$ and $\Delta T(CMIP'_{highS})$ broadly cover the range of $\Delta T(CMIP_{Full\_Future})$, suggesting that the bias and skill score are not good emergent constraints to reduce the uncertainty of $\Delta T$ in this study though the previous studies
have showed the reduction of their projection uncertainties (e.g. Smith and Chandler, 2010; Bracegirdle and Stephenson, 2013; Bracegirdle et al., 2013; Simpson et al., 2016). This is because the spatial pattern for the temperature is quite similar among the models and then the model selection using the score hardly has an impact on the reduction of uncertainty. On the other hand, the difference in the bias between the full set and the subset is large. The previous studies have suggested that the

performance of the present climate simulations is not necessarily related to the uncertainties of future projections (e.g., Knutti 2010, Shiogama et al. 2011) and we expected such a relation between the quantitative performance and the future change in this study.

The uncertainty range associated with the projected change in annual precipitation is shown in Fig. 3. Compared with $\Delta T$ in Fig. 2, model selection has a large impact on the reduction of the uncertainty in $\Delta P$, as was also found by MJ2016 using five GCMs used in the fast track of ISIMIP. The subsets of $\Delta P(CMIP'_{lowB})$ and $\Delta P(CMIP'_{highS})$ cover 70% and 60% of the full range of uncertainty from $CMIP_{Full\_Future}$ as the average over 14 regions, respectively, and cover the full range in Australasia (yellow and orange plots in Fig. 3). The largest difference between the coverages from $\Delta P(CMIP'_{lowB})$ and $\Delta P(CMIP'_{highS})$ appears in East Asia. Therefore, we need to pay attention that, when the model performance is the condition to select subsets, the uncertainty changes depending on which evaluation index are used, for example whether we use the bias or the skill score. The CORDEX subsets capture more than 50% of the full range in eight regions (Europe, MED, Africa, SEA, Australasia, Central America, South America and the Antarctica). On the other hand, the ISIMIP subsets capture less than 60% of the full range in all regions. In 11 regions, the CORDEX subsets capture a wider range than the ISIMIP subsets, a result markedly different than for $\Delta T$, where both CORDEX and ISIMIP have relatively large coverage as seen in Fig. 2. Therefore, the subset of four models used in ISIMIP2b has difficulty capturing the uncertainties in regional precipitation change. This result is the same as stated using the subset of five models used in the fast track of ISIMIP discussed by MJ2016, despite two of the five models changed.

The uncertainty range (maximum-minimum) is narrowed by using the subsets, but the interquartile range of $\Delta P(CORDEX)$, $IQR(\Delta P(CORDEX))$, shows a high coincidence with the $IQR(\Delta P(CMIP_{Full\_Future}))$, as well as with the $IQR(\Delta P(CMIP'_{lowB}))$ and $IQR(\Delta P(CMIP'_{highS}))$. The maximum−minimum range of $\Delta P(ISIMIP)$ also captures the $IQR(\Delta P(CMIP_{Full\_Future}))$. Therefore, the CORDEX and ISIMIP subsets can capture the average tendency of the change projected by the 25th to 75th percentile of $CMIP_{Full\_Future}$. In addition, the median of the uncertainty range is similar between the CORDEX subset and $CMIP_{Full\_Future}$. In Central Asia, the full range of $\Delta P(CORDEX)$ remains below the 25th percentile of $\Delta P(CMIP_{Full\_Future})$ while the maximum−minimum range of $\Delta P(ISIMIP)$ adequately covers the $IQR(\Delta P(CMIP_{Full\_Future}))$. Thus the three models of the CORDEX subset in Central Asia underestimate the average tendency of the change projected by $CMIP_{Full\_Future}$, despite that, differing from ISIMIP, CORDEX can select suitable models for discussion of climate change in Central Asia.

### 3.3 Comparison of uncertainty of the projected changes using randomly sampled models

We investigated whether the ISIMIP or CORDEX subsets were more suitable for capturing the uncertainty range obtained from $CMIP_{Full\_Future}$ by comparing the fractional coverage of uncertainty, FRA, of each subset with those of 10,000 randomly sampled subsets of $CMIP_{Full\_Future}$. As the result, the ISIMIP subset (four models) shows high coverage for the temperature change in all regions compared with the random samples and low coverage for the precipitation change in more than 60% of

all regions. By contrast, the CORDEX subset yields relatively wide coverage for the temperature and precipitation changes, but this depends on the number of models used.

Figure 4 illustrates FRA of the ISIMIP and CORDEX subsets (referred to $FRA_{ISIMIP}$ and $FRA_{CORDEX}$, respectively) in each region. Along the x-axis, the name of regions is arranged in ascending order of the number of models used in CORDEX. The number of models used in CORDEX is indicated in each parenthesis after the name, and by contrast, the number in ISIMIP is four in all regions. The y-axis indicates FRA of the uncertainty from each subset relative to that from the full set. The bar represents the distribution of the FRA values obtained from the possible 10,000 random samples ($FRA_{Random}$). The blue bar means the distribution using the subsets with four models as large as the ISIMIP subset ($FRA_{Random\_I}$), and the red bar means that with the same number of models used in CORDEX ($FRA_{Random\_C}$). Both ends of the bar indicate the lowest and highest values of FRA, and both ends of the bar with a dark color and horizontal line in the bar denotes the 25th and 75th percentiles and the median, respectively.

For the temperature change, $\Delta T$, $FRA_{ISIMIP}$ and $FRA_{CORDEX}$ (blue and red dots, respectively) exceed 60% in 13 and 10 regions, respectively (Fig. 4a). However, $FRA_{CORDEX}$ locates around the 25th percentile or less of $FRA_{Random\_C}$ (the bottom of dark red bar) in MED, East Asia, SEA, Europe, and the polar regions where $FRA_{CORDEX}$ is lower than $FRA_{ISIMIP}$. In the region with larger model ensemble in CORDEX, $FRA_{CORDEX}$ tends to be less than the median of $FRA_{Random\_C}$ (horizontal red line). On the other hand, $FRA_{ISIMIP}$ is typically around the 75th percentile (the top of dark blue bar) or higher than the median (horizontal blue line) of $FRA_{Random\_I}$ for all regions.

A relatively high coverage, above ~50%, is shown on $FRA_{CORDEX}$ for both changes of temperature and precipitation in eight regions when using nine models or more, except for temperature in Antarctica (Fig. 4a, b): that is to say, the CORDEX subset captures more than half of the range from $CMIP_{Full\_Future}$. The value of $FRA_{CORDEX}$ for $\Delta P$ is lower than that for $\Delta T$. A high coverage of more than 70%, however, can be gained by the CORDEX subset for $\Delta P$ in MED, South America, Europe, Australasia and Africa, which also indicates a high coverage compared with the median of $FRA_{Random\_C}$ (except for Europe) (Fig. 4b). In half of the regions, $FRA_{CORDEX}$ are in the range of the 25th percentile or less of $FRA_{Random\_C}$ (four regions of Asia, MENA, the Arctic, and North America). In Central and East Asia, and North America of these regions, $FRA_{CORDEX}$ is smaller than $FRA_{ISIMIP}$, even though CORDEX has the advantages of selecting suitable models for the region and also more models can be used, especially in East Asia and North America. The ISIMIP subsets in Antarctica and Australasia show a larger coverage than the 75th percentile of $FRA_{Random\_I}$, but the $FRA_{ISIMIP}$ of 60% is less than that for $\Delta T$. In more than 60% of all regions, $FRA_{ISIMIP}$ is less than the median of $FRA_{Random\_I}$; the averaged $FRA_{ISIMIP}$ over all regions is 33%.

From the FRA distributions estimated from the possible random samples regarding to both changes, $\Delta T$ and $\Delta P$, the IQR of $FRA_{Random\_C}$ itself rises toward a FRA of 100% as larger model ensemble are used. When random samples are composed of a subset with 15 models as large as subsets in CORDEX-Africa and -South Asia, the 75th percentile of $FRA_{Random\_C}$ is more than 90% in $\Delta T$ (Fig. 4a). In addition, the width of the IQR for $\Delta T$ is narrowed with increasing the number of models. The relationship between the number of models and FRA is clearly evident in $\Delta T$ because there is a small difference in $R_{Full}$ among regions for $\Delta T$ compared with $\Delta P$ (Fig. 2), and thus the larger model ensemble results in an increase in $FRA_{CORDEX}$ and

FRA$_{Random\_C}$. And also, we found that the probability of selecting model subsets with a low coverage was higher for precipitation than for temperature, even if the number of models selected increases.

From Fig. 4, the subsets with nine models or more can capture the uncertainty of projections in both temperature and precipitation widely, implying that there is a heterogeneity on the dataset by a different number of models (Gutowski et al.,
2016). We explored whether a similar tendency can be obtained in the other regions when the number of models changed. The same approach was performed by MJ2016. They estimated the coverages in each number of models to investigate the change of coverage performance of the subset with most widely covering the uncertainty in each of the global grid cells or the regional cells. On the other hand, in this study, to consider making better use of the current subsets, we investigated how the coverage changes with changing the number of models from the current model members.

Figure 5 shows the change of coverage performance with the number of models changing in each region. When the number of models is larger than the current number, we added models randomly selected to the current members. By contrast, when the number of models is less, we removed models randomly selected from the current members. Here we focused on the median of the FRA values obtained from the possible 10,000 random samples, meaning the FRA value obtained with a possibility of 50% when selected subsets randomly. For the temperature change, the median exceeds 60% in all regions when changing the
number of models from the current four ISIMIP members to seven members (Fig. 5a). The median above 60% is also obtained in 13 regions (except for Antarctica) when changing the number from the current CORDEX members to nine members. For the precipitation change, the coverage in nine members is above 50% in 10 regions and in 12 regions by changing the number of models from the current members in ISIMIP and CORDEX, respectively (Fig. 5b). Even when using nine members, the median is less than 50% in Four regions of MENA, Africa, and South and East Asia for the change of number from the ISIMIP
subset and in two regions of MENA and North America for that from the CORDEX subset.

The IQR for $\Delta$T shifts to a high FRA smoothly with the number of models in all regions. By contrast, the IQR for $\Delta$P sometimes gets large suddenly and/or shifts sharply, for instance, MENA and Africa. The discontinuous change is caused by a large variance of $\Delta$P from each model member. That is to say, when there are model members indicating a large change ratio relative to the other members, the coverage largely differs depending on the inclusion of the member with the large ratio or not. The
change amounts, $\Delta$T are similar among the model members and the variance is small. Thus, the FRA increases with the number of models and the IQR also increases smoothly. To prevent selecting the subset with a large change of the coverage depending on a model with extremely large or small change amount, investigating the variance of the projections in each region is needed when the number of models is decided.

## 4 Discussion

From the evaluation of the ability to reproduce the regional temperature and precipitation, it is found that the ISIMIP and CORDEX subsets include the models indicating a larger bias and a worse score than high performance models in the full set. Therefore, a much better model subset, with regard to biases and skill scores, can be selected with making use of the advantage of the small number of models. However, note that such a selection can be conducted when there are no constraints of data

availability which was the main constraint to select the current subsets in ISIMIP and CORDEX and when we use one variable of either temperature or precipitation. Focusing on one variable of either temperature or precipitation, 13 high performance models (out of 25) are included in both subsets of high-performance models for the bias and skill score (Supplement 3). In addition to the two indices of bias and skill score for one variable, the number of models indicating the high performance for

both two variables of temperature and precipitation is 0 at the minimum in Southeast Asia and the Arctic and 9 at the maximum in Africa. The averaged number over the regions is approximately 4. Therefore, for one variable, there is a possibility of 50% that a model with a small bias shows a high skill score but it is difficult to select such models for both variables of temperature and precipitation.

In this study, we assessed the current ISIMIP and CORDEX subsets to investigate whether their model ensemble indicates

small biases in the historical climatology and covers the uncertainty in the future projections widely using temperature and precipitation. Both variables are most frequently used in future projections and also weather forecasts. The evaluation for such a principal variable is important for the studies of ISIMIP and CORDEX. It should be noted, however, that ISIMIP needs a dataset with reasonable values for multiple variables used in their impact assessment and with enough coverage of the projection uncertainties. CORDEX requires the dataset with values based on a plausible mechanism of the climatology as the

input data for RCMs. Thus, it is possible that a good subset which we presented based on the model performance for temperature and precipitation will be an option of their future subsets.

Although ISIMIP and CORDEX have tight constraints for model selection at the present, both programs will select the subset showing a reasonable climate based on a plausible mechanism in the future. Two variables of temperature and precipitation are not possibly sufficient for the model selection. At least for the regional climatological studies and the assessment of its

impact, it is important to reproduce large-scale atmospheric circulation patterns which characterize the regional climate. Especially, the spatial pattern of precipitation depends on the accuracy of the circulation. Indeed, model change in ISIMIP from the fast track to ISIMIP2b has already been performed with a consideration of the ability to reproduce ENSO and monsoon (Frieler et al., 2017). The evaluation method used in this study can be applied to the other variables when we can obtain the reference data. For instance, Taylor's skill score which we used to evaluate the pattern of temperature and

precipitation can also apply to the pattern of circulation, indicated by sea level pressure (SLP) and geopotential height.

It is also preferable to select subsets in the next generations based on a combined approach that can consider the ability to reproduce multiple variables although, as described above, it is more difficult to obtain enough number of models as more variables and evaluation indices are employed. As the first step of the combined approach, it could be good that the evaluation of sea level pressure indicating the large-scale circulation which has an influence on the precipitation pattern, instead of

precipitation itself. This combination might obtain an adequate number of members, which is found to be difficult using the combination of temperature and precipitation here. Regarding combined approaches for future changes, Figure 4 presents that the coverage of $FRA_{Random\_C}$ is relatively high on both variables when the number of members is large. Thus, there would be a possibility to cover the projection uncertainties for both variables widely by applying a region-specific ensemble and an

adequate number of its ensemble members. The method above is one suggestion of the approaches and construction of such an approach would be one of the important tasks for both programs in the future.

## 5 Summary and conclusions

We explored the ability for the subsets of CMIP5 multimodel ensemble used in ISIMIP2b and CORDEX to reproduce the observed temperature and precipitation, and how the subsets capture the uncertainty in projected change of temperature and precipitation obtained from the full set of the ensemble. In addition, we discussed whether each subset shows a high coverage of the uncertainty in projected climate change compared with the possible subsets generated using 10,000 random samples. The spreads of the bias and Taylor's skill score from the subsets used in ISIMIP and CORDEX are smaller than those obtained from the full set of CMIP5 ensemble for the annual mean temperature and precipitation. However, despite of the smaller model ensemble in ISIMIP and CORDEX, the largest bias and the worst skill score distribute beyond the biases and the scores obtained from the half member subsets with less bias or high score of the full set. Therefore, although the ISIMIP and CORDEX approaches were able to select models that acceptably performed to represent the historical state, our results suggest that better subsets can be selected by focusing on smaller biases and/or higher scores for representing the historical climate. Note that such a selection can be performed when there are no constraints for the selection and when we use one variable of either temperature or precipitation as the evaluation index. A combined evaluation for both temperature and precipitation remains difficulty in obtaining an adequate number of models.

For the projected change in annual mean temperature, the subsets capture more than 60% of the uncertainty for the full set in the 13 terrestrial regions in ISIMIP and the 10 regions in CORDEX, from the total of 14 regions. The coverage of the uncertainty range by the ISIMIP subset is larger and equal to the coverage by the CORDEX subset in 10 regions by using only four models that are common to all regions. The FRA of the current CORDEX subset tends to be lower than the 50th percentile of the FRAs obtained from the possible 10,000 random samples in the regions where a large model ensemble is used. ISIMIP selected the subset of models with relatively high coverage of the uncertainty from the full set in all regions, compared with the 50th percentile from the random samples.

On the other hand, for the projected change in annual mean precipitation, the FRA for the CORDEX subset are around the 25th percentile or less of the FRAs from the random samples with the same number of models in half of all regions. However, CORDEX broadly captures the uncertainty range more than ISIMIP, differing from the temperature change. Additionally, a relatively high coverage (>50%) was obtained for the projections of both temperature and precipitation in eight regions when using nine models or more.

Compared with the random samples, the ISIMIP subset shows high coverage for the temperature change in all regions and, by contrast, low coverage for the precipitation change in more than 60% of the regions. The CORDEX subset is not performing well compared to the randomly selected samples but is marginally better than ISIMIP at covering uncertainties in the projected change in precipitation when a large model ensemble used. Therefore, the global common model set used in ISIMIP could have difficulty in capturing the uncertainty in regional precipitation change projections while capturing most of the uncertainty

in the temperature change projections. The region-specific model subset, like CORDEX, captures coverage of both uncertainties compared to the global common (ISIMIP) subset, but performs better when a large number of models is used. The current CORDEX subsets can capture both uncertainties for temperature and precipitation in the regions with a relatively large ensemble. However, it is found that changing the number of models from the current CORDEX members to nine members

can capture more than half of the full uncertainty in both projections of temperature and precipitation in more than 85% of all regions, with a possibility of 50%. Furthermore, the same is also shown as for the ISIMIP subset, but for 70% of all regions. Focusing on the uncertainty in the future projections, this result proposes that the current number of models need to be increased to seven, or nine models if possible, to discuss a similar uncertainty range among the regions.

In this study, we have assessed the subsets using the principal variables of temperature and precipitation. It is not sufficient for

selecting subsets in the next generations. We suggest that it is preferable a combined approach that can consider the ability not only for temperature and precipitation but also for the other ones which are also important to characterize the regional climate (e.g. the circulation patterns shown by sea level pressure and geopotential height). Construction of such an approach would be urgently demanded for both programs.

### Code and data availability

CMIP5 multimodel dataset is publicly available via the website of Earth System Grid Federation (http://pcmdi9.llnl.gov/). Observation products are publicly available online via each website: CRU (https://crudata.uea.ac.uk/cru/data/hrg/cru_ts_4.01/), CPC (https://ftp.cpc.ncep.noaa.gov/precip/CPC_UNI_PRCP/), GPCC (https://www.dwd.de/EN/ourservices/gpcc/gpcc.html), PRECL (http://ftp.cpc.ncep.noaa.gov/precip/50yr/gauge/), CMAP (https://ftp-cpc.ncep.noaa.gov/precip/cmap), GPCP (ftp://meso.gsfc.nasa.gov/pub/gpcp-v2.2/), MSWEP (http://www.gloh2o.org). Code for analysis is available to the editor and

reviewers for the purpose of the review. Public access to the code is limited due to the property of TOUGOU program, MEXT, Japan and, however, we can provide the code from the corresponding author upon request under the condition of collaborative research.

### Author contribution

All authors conceptualized the study and participated in the discussion. RI analysed the data and prepared the manuscript and

all authors revised the manuscript.

### Competing interests

The authors declare no conflict of interest.

### Acknowledgements

This work was conducted under the TOUGOU Program of the Ministry of Education, Culture, Sports, Science and Technology,

Japan and ERTDF 2-1904 of the Environmental Restoration and Conservation Agency, Japan. The authors acknowledge Dr

N. N. Ishizaki for useful suggestions and the anonymous reviewers for discussion and useful comments. All figures are created by the Generic Mapping Tools (GMT; http://gmt.soest.hawaii.edu) ver. 4.5.12.

**Supplement**

Supplement 1 is a list of the CMIP5 models used in CORDEX.

Supplement 2 describes the regional classification defined in CORDEX.

Supplement 3 describes the models with the top 50% of the CMIP5 models for the model bias and Taylor's skill score.

Supplement 4 describes the skill score for annual mean model precipitation over land.

Supplement 5 describes the annual mean model temperature bias over land.

Supplement 6 describes the skill score for the annual mean model temperature over land.

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

**Table 1: Number of CMIP5 models used in the CORDEX regions.**

| Region | | Region | |
|---|---|---|---|
| Europe | 13 | Southeast Asia (SEA) | 12 |
| Mediterranean (MED) | 5 | Australasia | 13 |
| Middle East and North Africa (MENA) | 5 | North America | 6 |
| Africa | 15 | Central America | 10 |
| Central Asia | 3 | South America | 9 |
| South Asia | 15 | Arctic | 5 |
| East Asia | 7 | Antarctica | 9 |

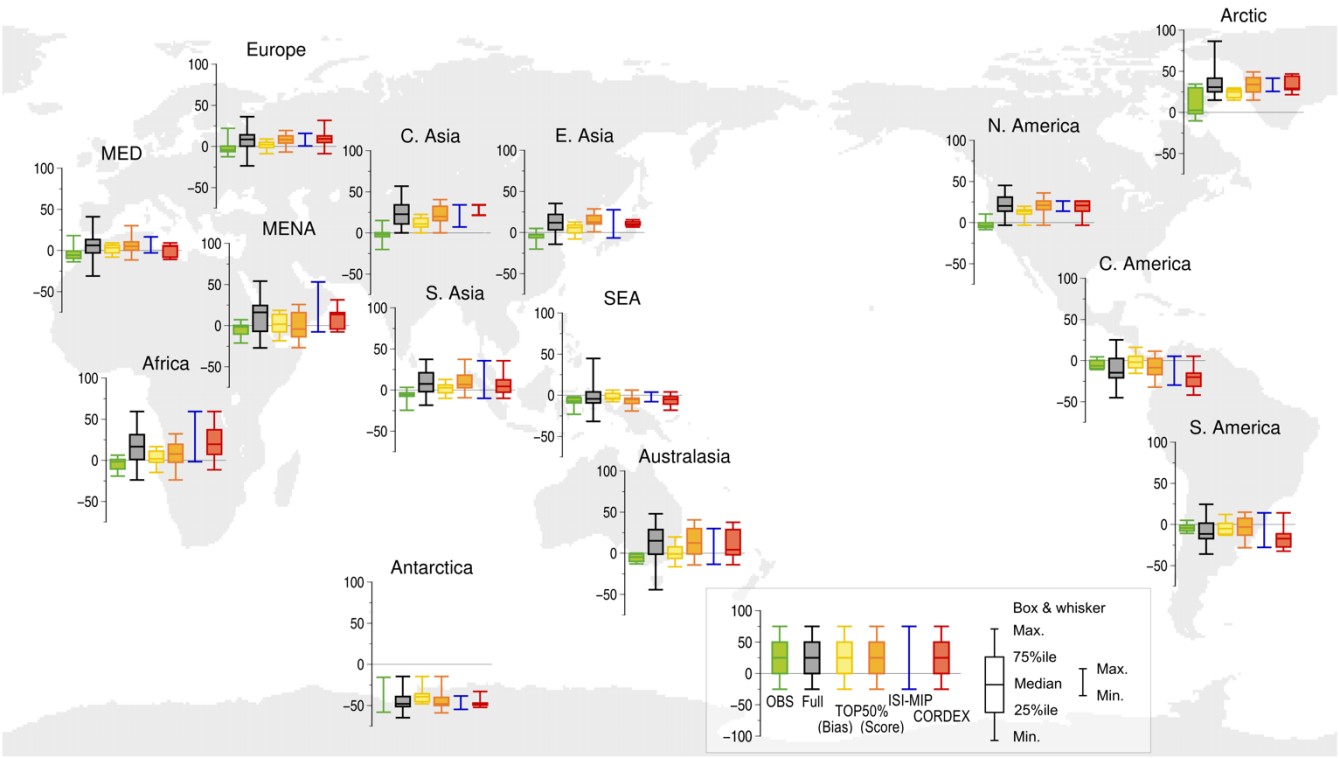

**Figure 1: Normalized annual mean model precipitation bias over land from the GPCC reference data (%). The bias was normalized by the regional average of GPCC data. The whiskers of the box plots show the range between the maximum and the minimum biases. The boxes and the lines within the boxes indicate the 25th to 75th percentile range and the median, respectively. Green plots indicate the deviations of six observation data from the reference data. The other plots indicate the model bias in the full set of 50 CMIP5 model set (black), the model sets with lower bias than the 50th percentile of biases of the full set (yellow), the model sets with higher Taylor's skill score than the 50th percentile of the scores of the full set (orange), and the model sets selected for ISIMIP (blue) and CORDEX (red).**

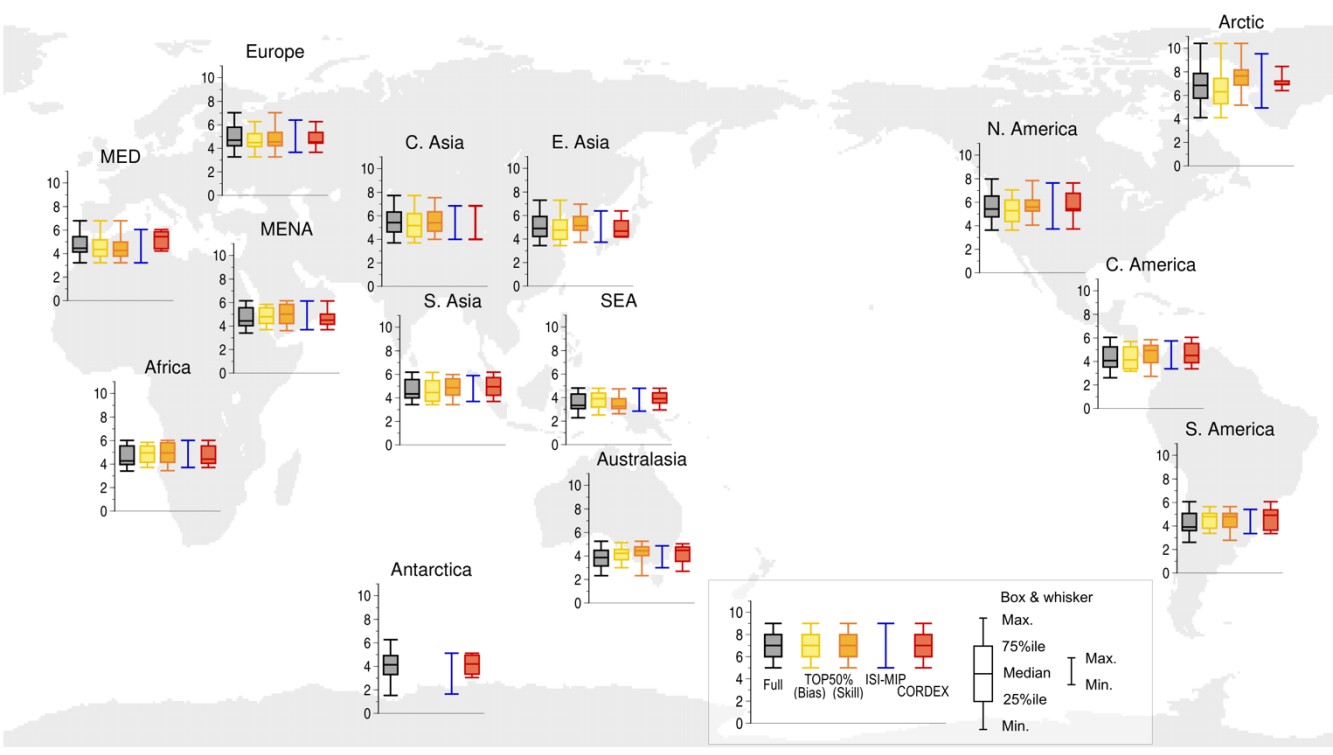

**Figure 2: Annual mean temperature increments in the future climate projection (K). The whiskers of the box plots show the range between the maximum and the minimum increments. The boxes and the lines within the boxes show the 25th to 75th percentile range and the median, respectively. Box plots indicate the increment in the full set of 42 CMIP5 models (black), the model sets with the top 50% of the CMIP5 models for the bias (yellow) or Taylor's skill score (orange), and the model sets selected for ISIMIP (blue) and CORDEX (red). The top 50% of the CMIP5 models cannot be plotted over Antarctica because of missing the CRU reference data.**

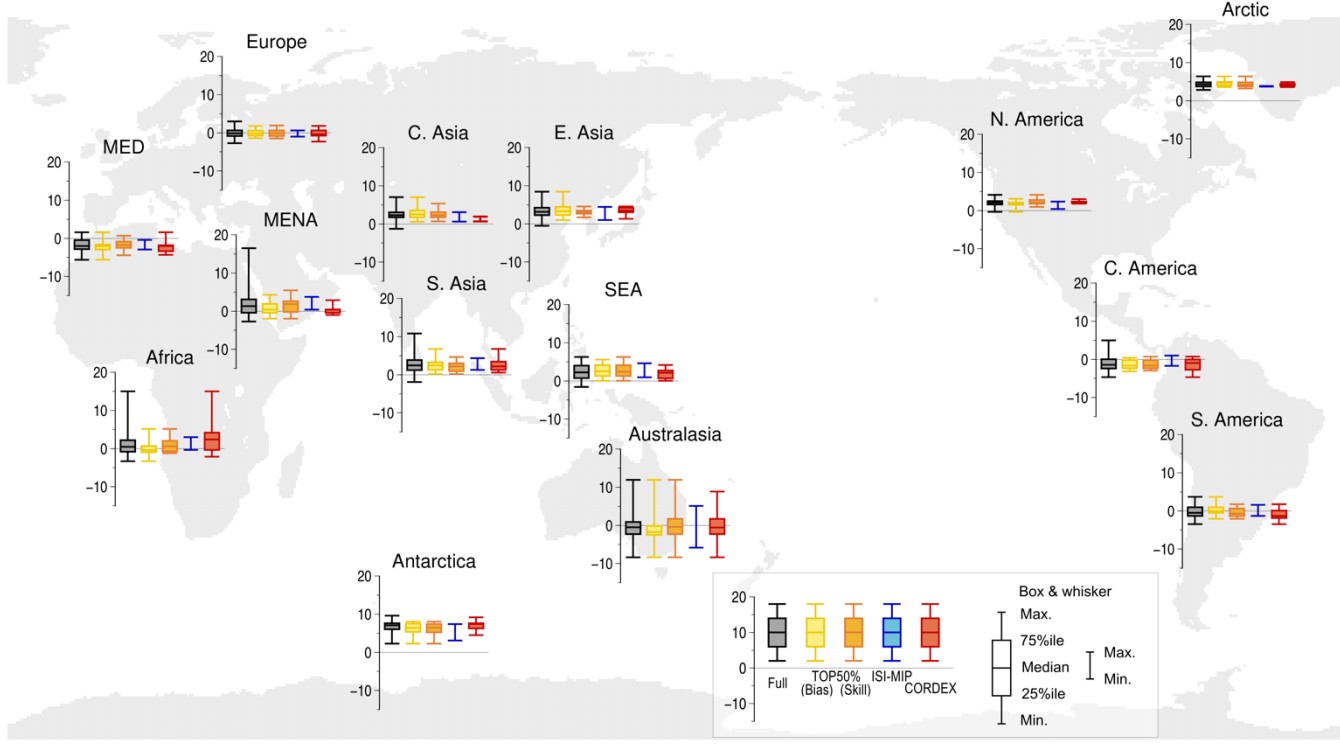

**Figure 3: As for Figure 2, but for the projected change in annual mean precipitation scaled to the regional mean temperature increment over the land (% K⁻¹).**

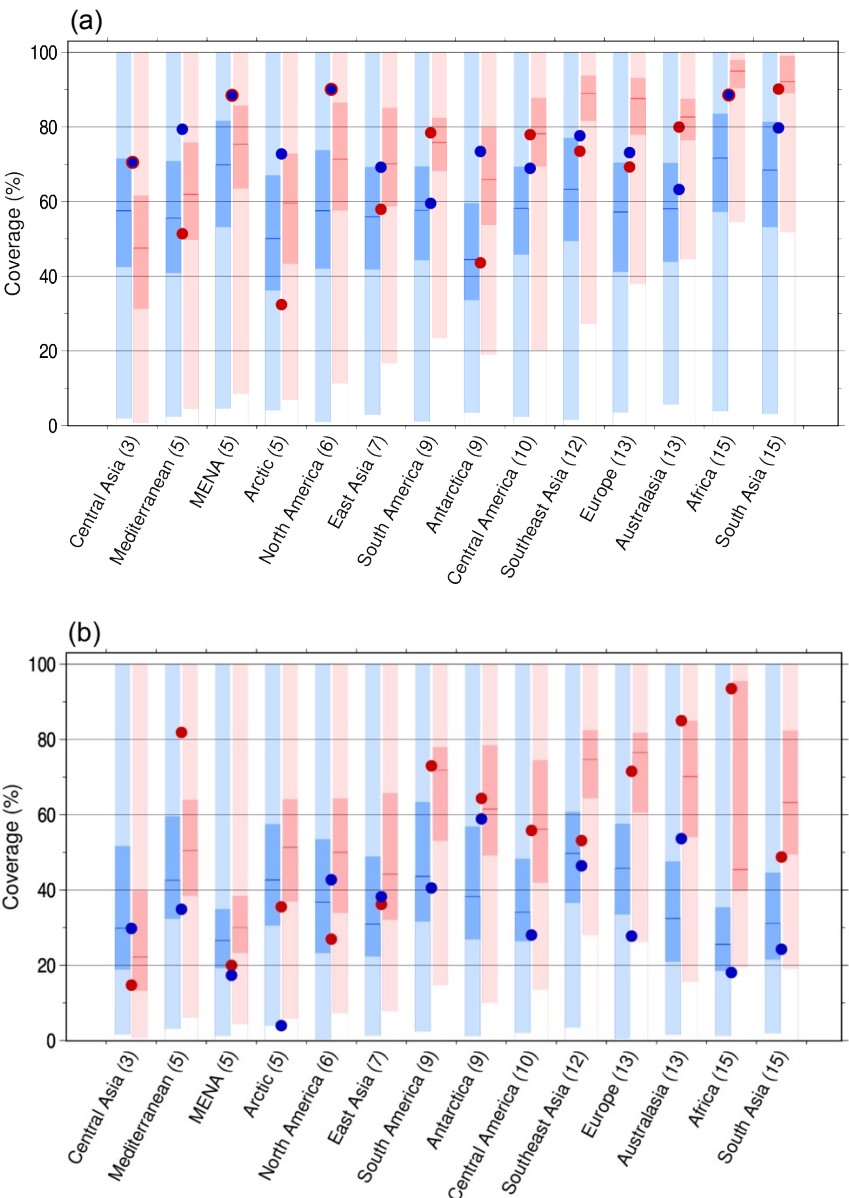

**Figure 4: Coverage performance of the ISIMIP and CORDEX subsets compared with the range of the full set of CMIP5 models for (a) annual mean temperature increment and (b) precipitation change scaled to the regional mean temperature increment. Blue and red dots indicate the coverage (FRA) in ISIMIP and CORDEX, respectively for each region. Blue bars indicate the spread of FRA when four models, as in ISIMIP, are selected randomly in 10,000 times. Red bars indicate the spread when randomly selecting the same number of models as in CORDEX; e.g., 10 models in Central America. The full range of the colored bars indicates the minimum to maximum coverage. Dark blue and red bars indicate the 25th to 75th percentile range of the FRA spread. Horizontal lines in the dark blue and red regions indicate the median. Numbers in parentheses are the number of models used in CORDEX.**

(a)

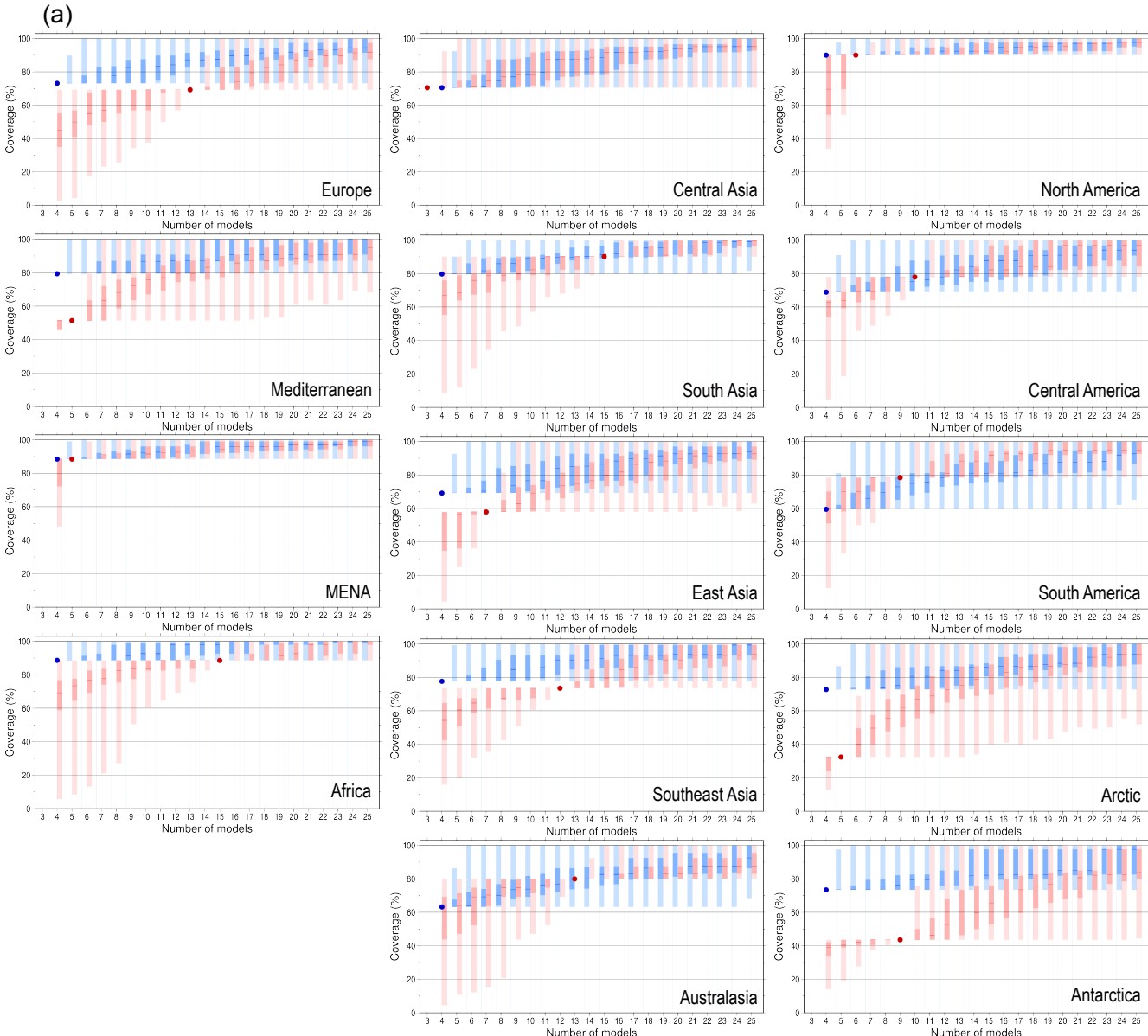

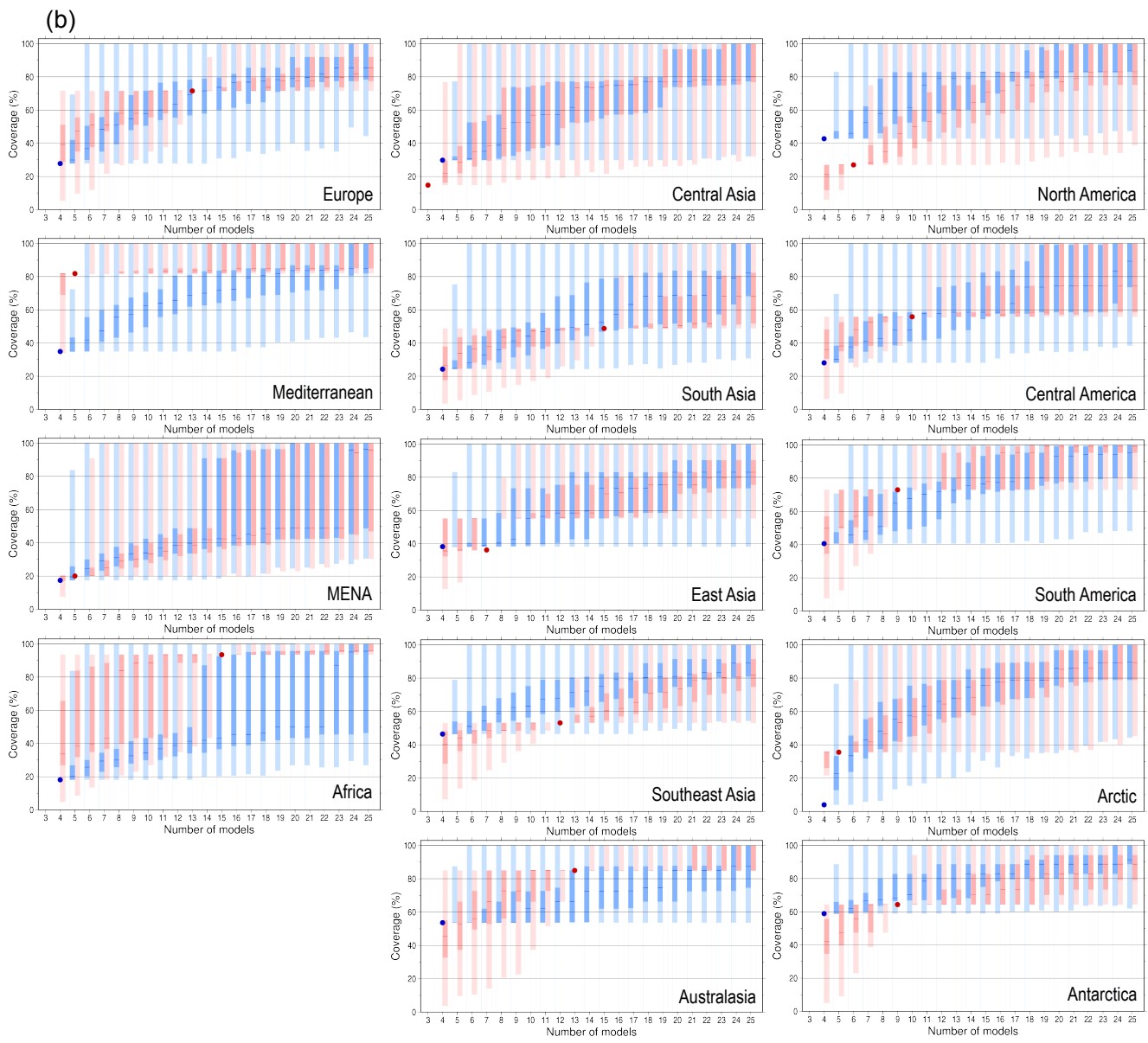

**Figure 5: Change of coverage performance of the ISIMIP and CORDEX subsets depending on the numbers of selected models in each region for (a) annual mean temperature increment and (b) precipitation change scaled to the regional mean temperature increment. As in Fig. 4 but the x-axis denotes the number of selected models.**

