# Peer review of "Uncertainties in climate change projections covered by the ISIMIP and CORDEX model subsets from CMIP5"

_Geoscientific Model Development, 2019_

## Referee Comment (RC1) · Anonymous Referee #1 · 23 Aug 2019

Summary and General comments

The paper by Ito et al. investigates the uncertainty ranges in projections from the ISIMIP and CORDEX projects. Both of these projects selected a sub-sample from CMIP5 Global Climate Models (GCMs) to bias correct and then drive impact models (ISIMIP) or to downscale the GCM's (CORDEX). ISIMIP and CORDEX have different goals and also the number of models selected and the approach to sub-select the GCMs were different. The authors look into how well these two projects cover the uncertainty ranges provided by the original CMIP5 model set. They show that the ISIMIP and CORDEX uncertainty ranges are smaller than the original range but still larger

than from a subset only selecting well performing models, even though the number of models selected in ISIMIP and CORDEX was smaller than the number of well performing models they were compared to. The authors also conclude that better subsets with smaller biases and/or higher scores would be possible than the current ISIMIP and CORDEX selections.

While it is interesting to see how different the uncertainty ranges of different model selections are, I am not necessarily sure if the comparison is fair, given that as far as I know neither ISIMIP nor CORDEX selected their GCMs based on these criteria. Among other points explained below, I am also missing a clear recommendation that would help the next rounds of ISIMIP and CORDEX to sub-select their GCMs.

Specific comments

For ISIMIP the main constraint in choosing GCMs was data availability, and they needed many more variables than the ones the authors consider in this study. Hence, even if "better" subsets in terms of performance based on precipitation and temperature would be possible, that does not necessary mean these subsets would have been an option for the ISIMIP project. For CORDEX data availability was also a major constraint, so again, even if better subsets based on temperature and precipitation would have been possible, if the data to drive the RCMs was not available that would not have helped the CORDEX project. These aspects should at least be discussed in the manuscript.

I was also missing the link from the performance in the historical projections to the projected uncertainty ranges. Do the sub-sampling based on lower bias/higher score cover larger, smaller or similar uncertainty ranges in the projections? The data is all there in the figures, but it is not discussed in the text.

I also find it hard to believe that neither the ISIMIP nor the different CORDEX regions did any analysis similar to what the authors provide here? At least for ISIMIP Mc-Sweeney and Jones (2016) seem to already have done this in a very comprehensive

way. What is this study adding on top of that?

On page 8, lines 19-22, the authors mention results what would happen if a larger number of models would have been used in the Central Asia region. This result, I imagine something similar to Figure 3 in McSweeney and Jones (2016) but for the CORDEX regions, would have been very interesting. I think it would allow to show how many models would need to be selected to cover a certain uncertainty range, which would help to make a recommendation for the next round of CORDEX. I would also be curious to see if these numbers differ between different regions.

Technical corrections

Figures: While I kind of like the illustration of the graphs on the map it takes up quite a lot of space while the graphs itself are rather small. I wonder if the graphs could be increased but would take up less space in a more classical arrangement?

Supplement 1: I find this table not very informative, I would be more interested to know in which regions which models were used than in how many regions each model was used.

Supplement 4 and 5: I think the Obs are missing in these Figures.
* * *

---

## Referee Comment (RC2) · Anonymous Referee #2 · 5 Sep 2019

General comments

This manuscript aims to quantify the spread of CMIP5 projections and biases covered by the subsets of models used in the ISIMIP and CORDEX experiments. The first section of the results examines the spread of model performance in reproducing the temperature and precipitation over the historical period (1986-2005), relative to a range of observational and reanalysis products. The rest of the results examines the spread in projected end-of-21st-century changes in annual mean temperature and precipitation, and how it compares to the spread covered by randomly selected subsets. The main findings are that (i) the small ensembles used in ISIMIP and CORDEX generally

perform well over the historical period but are not optimal in minimizing historical biases, and (ii) the ISIMIP ensemble outperforms the CORDEX and randomly selected ensembles in covering the full CMIP5 range of projected temperature changes, but both ISIMIP and CORDEX cover a smaller spread of precipitation changes than randomly selected subsets.

This manuscript presents a valuable study to put the CORDEX and ISIMIP subsets in the context of the full CMIP5 ensemble. At this stage, it is mostly descriptive and would greatly benefit from a more comprehensive discussion, including the benefits/limitations of the metrics used, and the implications of its findings. Please clarify how this specific study sets itself apart from existing studies such as McSweeney & Jones (2016), and how your results fit into the context of the existing literature. Minor adjustments to language and sentence structure are needed to improve the readability of the manuscript.

Specific comments

Section 1 Introduction P2 L. 19: Please specify what these previous studies have found, and why we have yet to reach a consensus on the method to select small ensembles. Also, revisiting these papers in a discussion section would provide the necessary context to interpret the results and whether the methodology used in this study is distinct from, or improves upon, the ones used in these studies.

P3 L. 23-36 Please clarify what is specific to this study: is some aspect of the methodology new? Is this performing an existing analysis to a new set of data? Is the added value of the manuscript to specifically address whether region-specific subsets (CORDEX) outperform a globally consistent sub-set (ISIMIP)? Stating this explicitly would improve the value and readability of the manuscript.

Section 2.1 P4 L. 18 Why focus on land area only? Regional precipitation, including over the seas/ocean, is relevant for impact studies. Please clearly state the scope (and the application) of this study.

P4 L. 22 Can you justify why you excluded low-precipitation models from the precipitation analysis? I understand these models significantly bias your ensemble but this undermines the stated aim of the study (i.e. to quantify the spread of the full CMIP5 ensemble covered by the ISIMIP and CORDEX subsets). You arbitrarily reduced the model spread covered in this study. Please at least provide more information as to how many models were excluded (and thus the size of your remaining ensemble), why 0.1mm/day was chosen as a threshold, and the reasoning to exclude these models from the precipitation study but to keep them for temperature (if the argument is that the climate they produce is too unrealistic to be a plausible representation of today's climate).

P4 L. 32 Please include a definition of the skill score used here, or a reference to a published paper using the exact same skill score. In Taylor (2001), two examples of skill scores are used, to illustrate that the skill score can be adjusted depending on whether you value high correlation or matching the variability most. In addition, it is explicitly stated that the value of Ro should be reported every time a skill score is used.

P5 L5 Is R the min-max range of a given subset? Please clarify. Using 'uncertainty range' is misleading; it sounds like you are sampling your ensemble. If I understand correctly, you generate 10,000 values of RSub, then look at ensemble spread (in Fig 4).

Section 3 This section is entitled 'Results and Discussion' but mostly contains the description of the results. Regardless of whether it is included in this section or a separate section, the manuscript needs a more comprehensive discussion (see other comments below).

Section 3.1 P.5, L. 28-29 Please include the top 50% ensembles in the supplementary material, so that future model selection can rely on your analysis to select less biased models.

P5 L29-30 Can you suggest why the spread is different between the two ensembles

over the Northern Hemisphere? In Fig.1, the spread for ISIMIP is sometimes significantly larger, sometimes significantly smaller than CORDEX. Could this be due to the number of models in CORDEX? Do some of the regions have models that overlap across ISIMIP and CORDEX? Simply stating that they are different in some regions is not very informative.

P6 L1-3 This paragraph would benefit from an earlier explanation about how the skill and bias metrics are different and the insight gained by using both. Include some interpretation of why the model ensembles that perform relatively well in a bias metric perform less well in the skill metric. (same comment for P6 L11)

P6, L. 15-16 Please include the top 50% ensembles in the supplementary material. In addition, please include in the discussion whether the 'best performing models' perform well both in temperature and precipitation, and whether selecting according to high skill or low bias makes a difference. As you state that a better ensemble can be selected, please give the evidence from your results that this can be done robustly.

Section 3.2 P6, L.25 Please place this into context by mentioning other studies that have looked at emergent constraints, even if it's only in specific regions (e.g. Bracegirdle et al, 2018 for Southern Ocean winds; Bracegirdle and Stephenson 2013 for Arctic warming).

Section 3.3

In general, this section is confusing. It would benefit from clearly stating what is being compared, and referring to specific aspects of Fig 4 to support your statements. Specifically: P8 L1-2: Please clarify which metric you use to make that statement (i.e. the total coverage on the y-axis). I got confused because the performance of FRACORDEX remains low compared to FRARandom_C , even as the number of models increases. Please state explicitly where you are comparing it to the full range, or to Random_C (3 sentences later). P8 L16: Please specify which FRA you are talking about: the median of FRARandom_C,? Or FRACORDEX? Or both? P8 L19-22: This

is an interesting point, but if you make the point that increasing the number of models produces a higher FRA, please show the evidence for it. The latter part of the description is unclear so adding the technical details and a figure would make a stronger point.

Summary and Conclusions

This section provides a general summary of the findings, but would benefit from providing context as to how these results compared to other studies (e.g. those cited in the introduction), and how the findings advance the general understanding of the field. In addition, statements in P9 L4-5 and P9 L 15-16 seems to indicate CORDEX performance to be bad relative to randomly selected ensembles, while P9 L8-9 states 'relatively wide coverage of both uncertainties'. Please clarify so that the message is not ambiguous. For example, it is ok to state that CORDEX is not performing well compared to the randomly selected ensembles, but is marginally better than ISIMIP at sampling uncertainties in projected change in precipitation.

Please include a more comprehensive discussion of your methods and results, including:

Two metrics for "good performance" are used in parallel throughout the study (low bias and high skill score). Please comment as to how similar/distinct these two metrics are, and on the insights gained by using both (qualitatively or quantitatively). Similarly, how different are the 'top 50%' ensembles? i.e. does using skill or bias for selection of the best performing models significantly affect the ensemble?

The results section 3.1 focuses mostly on whether the ISIMIP and CORDEX fall within the observational spread (e.g. L. 2-3 on page 6). It would be helpful to distinguish whether this is mainly due to a large spread in the model ensembles, or whether a systematic bias is seen in certain regions (e.g. Fig 1 shows model ensembles overestimate precipitation in most regions). Also, please include a discussion of the expected variance of model ensembles. In coupled models, the timing of climate variability modes

is unlikely to match that of observations, so the variance over a 20-year period is likely to be higher in model ensembles than observations.

In this study, the performance of CORDEX and ISIMIP are considered independently for the temperature and precipitation changes (with precipitation being scaled by the temperature change). Please discuss whether there is any evidence that a selection on one variable (e.g. precipitation) is sufficient to select good performing models, or whether a combined approach is necessary to select models. In climate impacts, people care about the plausibility and diversity of climate sampled, not a single variable.

Technical corrections

P1 L18 (and after) High performed models -> high performance models or high-fidelity models

P1 L20-25 Please rework this section to clarify the meaning. As you have not previously introduced the 10,000 sampling strategy, these two sentences are confusing.

P2 L33 Please rephrase this sentence for better readability. For example: "In addition, paper X and Y showed that combining region-specific subsets covers more uncertainty than a single, globally consistent, subset of models."

P5, L11 Please rephrase that last sentence for readability.

P7, L9-12 This sentence needs to be reworked for readability.

P7, L15 "with those of the 10,000" -> remove "the"

P7 L16 "randomly sampled subsets" of what?

P9 L17 Be more specific: 'it depends on the number of models used' is too vague to be informative -> "FRA increases with the number of models used", or "regions covered by bigger ensembles generally have higher FRA"...

P13 L8 "areal mean of the reference data" -> normalized by the regional average of

[Figure]

GPCC data.

P14 Figure 2 Why does Antarctica have no top 50% in temperature? Explain that somewhere (main text or figure caption).

P16 Figure 4 "uncertainty range" -> range Also, why are red dots missing in some regions in Fig 4a?
* * *

---

## Author Comment (AC1) · 11 Oct 2019

Response to the comments from Anonymous Referee #1 for the manuscript:
"Uncertainties in climate change projections covered by the ISIMIP and CORDEX model subsets from CMIP5" by Ito et al.

We would like to appreciate your careful review and constructive comments and suggestions for improving our manuscript. We almost agree with them. We have made modifications through our manuscript according to the responses. Please check our detailed responses below. The numbers of page and line are corresponding to the number in the original file (https://www.geosci-model-dev-discuss.net/gmd-2019-143/gmd-2019-143.pdf).

In this modification, we added a CMIP5 model of CSIRO-Mk3L-1-2, to the original 49 models for the historical run we analyzed. It is because there is no member of r1i1p1 by CSIRO-Mk3L-1-2, but there is r1i2p1 as well as CESM1-WACCM which was already used. The results did not change from the original manuscript by this modification. I apologize for the change.

In this revision, McSweeney and Jones (2016) have referred to as MJ2016 except for the first reference.

Thank you once again for your review.
We would be glad to respond to any further comments you may have.

--- **Summary and General comments**

The paper by Ito et al. investigates the uncertainty ranges in projections from the ISIMIP and CORDEX projects. Both of these projects selected a sub-sample from CMIP5 Global Climate Models (GCMs) to bias correct and then drive impact models (ISIMIP) or to downscale the GCM's (CORDEX). ISIMIP and CORDEX have different goals and also the number of models selected and the approach to sub-select the GCMs were different. The authors look into how well these two projects cover the uncertainty ranges provided by the original CMIP5 model set. They show that the ISIMIP and CORDEX uncertainty ranges are smaller than the original range but still larger than from a subset only selecting well performing models, even though the number of models selected in ISIMIP and CORDEX was smaller than the number of well performing models they were compared to. The authors also conclude that better subsets with smaller biases and/or higher scores would be possible than the current ISIMIP and CORDEX selections.

While it is interesting to see how different the uncertainty ranges of different model selections are, I am not necessarily sure if the comparison is fair, given that as far as I know neither ISIMIP nor CORDEX selected their GCMs based on these criteria. Among other points explained below, I am also missing a clear recommendation that would help the next rounds of ISIMIP and CORDEX to sub-select their GCMs.

We glad to hear your interests in our study. We have made our response to the comments about the unfair comparison between the subsets from ISIMIP and CORDEX and about the recommendations towards the next rounds, as the responses to specific comment #1 and #4 respectively. Please find below.

--- **Specific comments**
1. For ISIMIP the main constraint in choosing GCMs was data availability, and they needed many more variables than the ones the authors consider in this study. Hence, even if "better" subsets in terms of performance based on precipitation and temperature would be possible, that does not necessary mean these subsets would have been an option for the ISIMIP project. For CORDEX data availability was also a major constraint, so again, even if better subsets based on temperature and precipitation would have been possible, if the data to drive the RCMs was not available that would not have helped the CORDEX project. These aspects should at least be discussed in the manuscript.

We appreciate your accurate comments. Our explanation was not sufficient. The "better subset" is based only on the model bias and Taylor's skill score in our analysis. From an additional analysis in this revision, it is found that such a subset can be obtained under the condition without considering the data availability and with focusing on one variable of temperature or precipitation. We have described the following sentence to the section of discussion which made in this revision.

(P8 L23) "… a much better model subset, regarding to biases and skill scores, can be selected with making use of the advantage of the small number of models. However, such a selection can be conducted when there are no constraints of data availability which was the main constraint to select the current subsets in ISIMIP and CORDEX and when we use one variable of either temperature or precipitation."

As you noted, ISIMIP and CORDEX select their subset under the different constraints at the present. We have also added the followings in the section of discussion:

(P8 L23) "In this study, we assessed the current ISIMIP and CORDEX subsets to investigate whether the subset indicates small biases in the historical climatology and covers the uncertainty in the future projections widely using temperature and precipitation. Both variables are most frequently used in future projections and also weather forecasts. The evaluation for such a principal variable is important for the studies of ISIMIP and CORDEX. It should be noted, however, that ISIMIP needs the dataset with reasonable for multiple variables used in their impact assessment and with enable to discuss the uncertainty in the projections. CORDEX requires the dataset with based on a plausible mechanism of the climatology as the input data for RCMs. Thus, there is a possibility that a good subset which we presented based on the model performance for temperature and precipitation will be an option of their future subsets."

2.  I was also missing the link from the performance in the historical projections to the projected uncertainty ranges. Do the sub-sampling based on lower bias/higher score cover larger, smaller or similar uncertainty ranges in the projections? The data is all there in the figures, but it is not discussed in the text.
    We had mentioned the uncertainty range for the temperature change obtained from the subsets on P6, L24-26 and on the other hand, for the precipitation change on P6, L30. Especially for precipitation, there was less explanation. We have added the description below to P6, L29. With the addition, we have modified a whole of the paragraph more understandably.

"The subsets of $\Delta P(CMIP'_{lowB})$ and $\Delta P(CMIP'_{highS})$ cover 70% and 60% of the full range of uncertainty from $CMIP_{Full\_Future}$ as the average over 14 regions, respectively, with totally covering the full range in Australasia. The largest difference between the coverages from $\Delta P(CMIP'_{lowB})$ and $\Delta P(CMIP'_{highS})$ appears in East Asia. Therefore, we need to pay attention that, when the model performance is the condition to select subsets, the uncertainty changes depending on which evaluation index are used, like at least the bias or the skill score."

3.  I also find it hard to believe that neither the ISIMIP nor the different CORDEX regions did any analysis similar to what the authors provide here? At least for ISIMIP McSweeney and Jones (2016) seem to already have done this in a very comprehensive way. What is this study adding on top of that?
    McSweeney and Jones (2016) (hereafter MJ2016) have discussed the uncertainty in the projections but not mentioned the ability to represent the present-day climate and the projections itself which we have investigated. Also, as the update from MJ2016, we have analysed four GCMs used in the newer round of ISIMIP, instead of the GCMs analysed in MJ2016. On the other hand, as you pointed out, there are some CORDEX regions where their GCM subsets have been assessed but the assessments are limited.
    Uniform assessment over the regions permits to discuss the difference of performance among the regions. In addition, Gutowski et al. (2016) have mentioned there is a possibility of the heterogeneity on climate information among the regions as one of the main problems in CORDEX. This study has indicated that the subsets can widely capture the uncertainty in both projections of temperature and precipitation in the regions with a large ensemble. Thus, it is found the heterogeneity exists in the current dataset when focusing on the uncertainty. Furthermore, from the added results in this revision, we suggest that nine model members are needed to solve the heterogeneity of the uncertainty.
    From the assessment of the subsets selected in each program in the same method, we understand how different the climate information from a global consistent subset is from the original one by using the ISIMIP subset in the CORDEX framework, with assuming CORDEX CORE.
    We have added the above contents to P4 L3.

"The ability for the ISIMIP subset was not mentioned by MJ2016 and thus we investigated that in region by region. We analysed four GCMs selected in ISIMIP2b (unless specified otherwise, hereafter refers to as ISIMIP) here. Thus, assessment of the projections was also updated from MJ2016. The GCMs used in CORDEX have been assessed by region in previous studies, but are limited (e.g., Haensler et al. 2013 for Africa; Bartók et al. 2017 for Europe; Karmalkar 2018 for North America). Even simple assessment conducted is needed for the present CORDEX. Furthermore, uniform assessment across regions permits to discuss the difference of characteristics among the regions and the possibility of heterogeneous scenario as mentioned above. By using the subsets from the two programs, we can explore the difference between the original subset in CORDEX and the subset selected with assuming CORDEX CORE, which is helpful information for the model selection in CORDEX CORE."

4. On page 8, lines 19-22, the authors mention results what would happen if a larger number of models would have been used in the Central Asia region. This result, I imagine something similar to Figure 3 in McSweeney and Jones (2016) but for the CORDEX regions, would have been very interesting. I think it would allow to show how many models would need to be selected to cover a certain uncertainty range, which would help to make a recommendation for the next round of CORDEX. I would also be curious to see if these numbers differ between different regions.

We appreciate your constructive suggestion to gain more insight into our results. We added the results about the change of coverage depending on the number of models in each region to Section 3.3. We have referred the idea by McSweeney and Jones (2016). They have changed the number of models to explore how the coverage changes with the number of models when a subset covers the uncertainty in each grid most widely over the globe or regions. On the other hand, in this study, to consider making better use of the current subsets, we have changed the number of models from the current model members and explored how the coverage changes. The details are as what followings:

(P8 L19) "From Fig. 4, the subsets with nine models can capture the uncertainty of projections in both temperature and precipitation widely, implying that there is a heterogeneity on the dataset by a different number of models (Gutowski et al. 2016). We explored whether a similar tendency can be obtained in the other regions when the number of models changed. The same approach was performed by MJ2016. They focused on a subset covering the uncertainty in each grid most widely over the globe or regions and investigated how the coverage changes with the number of models. On the other hand, in this study, to consider making better use of the current subsets, we investigated how the coverage changes with changing the number of models from the current model members.

Figure 5 shows the change of coverage performance with the number of models changing in each region. When the number of models is larger than the current number, we added models randomly selected to the current members. By contrast, when the number of models is less, we removed models randomly selected from the current members. Here we focused on the median of the FRA values obtained from the possible 10,000 random samples, meaning the FRA value obtained with a possibility of 50% when selected subsets randomly. For the temperature change, the median exceeds 60% in all regions when changing the number of models from the current four ISIMIP members to seven members which are less than nine members (Fig. 5a). The median above 60% is also obtained in 13 regions (except for Antarctica) when changing the number from the current CORDEX members to nine members. For the precipitation change, the coverage in nine members is above 50% in 10 regions and in 12 regions by changing the number of models from the current members in ISIMIP and CORDEX, respectively (Fig. 5b). Even when using nine members, the median is less than 50% in Four regions of MENA, Africa, and South and East Asia for the change of number from the ISIMIP subset and in two regions of MENA and North America for that from the CORDEX subset.

The IQR for ΔT shifts to a high FRA smoothly with the number of models in all regions. By contrast, the IQR for ΔP sometimes gets large suddenly and/or shifts sharply, for instance, MENA and Africa. The discontinuous change is caused by a large variance of ΔP from each model member. That is to say, when there are model members indicating a large change ratio relative to the other members, the coverage largely differs depending on the inclusion of the member with the large ratio or not. The change amounts, ΔT are similar among the model members and the variance is small. Thus, the FRA increases with the number of models and the IQR also increases smoothly. To prevent selecting the subset with a

large change of the coverage depending on a model with extremely large or small change amount, investigating the variance of the projections in each region is needed when the number of models is decided."

(P9 L18) "The current CORDEX subsets can capture both uncertainties for temperature and precipitation in the regions with a relatively large ensemble. However, it is found that changing the number of models from the current CORDEX members to nine members can capture more than half of the full uncertainty in both projections of temperature and precipitation in more than 85% of all regions, with a possibility of 50%. Furthermore, the same is also shown as for the ISIMIP subset, but for 70% of all regions. Focusing the uncertainty in the future projections, this result proposes that the current number of models need to be changed to discuss a similar uncertainty range among the regions."

**--- Technical corrections**
1. Figures: While I kind of like the illustration of the graphs on the map it takes up quite a lot of space while the graphs itself are rather small. I wonder if the graphs could be increased but would take up less space in a more classical arrangement?

We appreciate your suggestion. We can understand a lot of space, especially in Supplement 4 and 5. We deeply considered the modification but the graphs on the map is good from the point of seeing the property corresponding to the region at a glance. We have redrawn the figures with reducing the space as much as possible.

2. Supplement 1: I find this table not very informative, I would be more interested to know in which regions which models were used than in how many regions each model was used.

The table has been changed to a table presenting the models used in each CORDEX regions. Please check the modified manuscript.

3. Supplement 4 and 5: I think the Obs are missing in these Figures.

Differed with the precipitation, one observation dataset, CRU, is used as the temperature reference data as indicated on P4, L26-27. Thus, there is no plot for the observation.
* * *
We have revised our manuscript to address comments from Anonymous Reviewer #1.

---

## Author Comment (AC2) · 11 Oct 2019

Response to the comments from Anonymous Referee #2 for the manuscript:
"Uncertainties in climate change projections covered by the ISIMIP and CORDEX model subsets from CMIP5" by Ito et al.

We would like to appreciate your careful review and constructive comments and suggestions for improving our manuscript. We almost agree with them. We have made modifications through our manuscript according to the responses. Please check our detailed responses below. The numbers of page and line are corresponding to the number in the original file (https://www.geosci-model-dev-discuss.net/gmd-2019-143/gmd-2019-143.pdf).

In this modification, we added a CMIP5 model of CSIRO-Mk3L-1-2, to the original 49 models for the historical run we analyzed. It is because there is no member of r1i1p1 by CSIRO-Mk3L-1-2, but there is r1i2p1 as well as CESM1-WACCM which was already used. The results did not change from the original manuscript by this modification. I apologize for the change.

In this revision, McSweeney and Jones (2016) have referred to as MJ2016 except for the first reference.

Thank you once again for your review.
We would be glad to respond to any further comments you may have.

**--- General comments**
This manuscript aims to quantify the spread of CMIP5 projections and biases covered by the subsets of models used in the ISIMIP and CORDEX experiments. The first section of the results examines the spread of model performance in reproducing the temperature and precipitation over the historical period (1986-2005), relative to a range of observational and reanalysis products. The rest of the results examines the spread in projected end-of-21st-century changes in annual mean temperature and precipitation, and how it compares to the spread covered by randomly selected subsets. The main findings are that (i) the small ensembles used in ISIMIP and CORDEX generally perform well over the historical period but are not optimal in minimizing historical biases, and (ii) the ISIMIP ensemble outperforms the CORDEX and randomly selected ensembles in covering the full CMIP5 range of projected temperature changes, but both ISIMIP and CORDEX cover a smaller spread of precipitation changes than randomly selected subsets.

This manuscript presents a valuable study to put the CORDEX and ISIMIP subsets in the context of the full CMIP5 ensemble. At this stage, it is mostly descriptive and would greatly benefit from a more comprehensive discussion, including the benefits/limitations of the metrics used, and the implications of its findings. Please clarify how this specific study sets itself apart from existing studies such as McSweeney & Jones (2016), and how your results fit into the context of the existing literature. Minor adjustments to language and sentence structure are needed to improve the readability of the manuscript.

We appreciate your useful comments to improve our manuscript. We have made our response to the general comments as the response to the following specific comments:
- the benefits/limitations of the metrics used and the implications of its findings (#11, #15)
- how this specific study sets itself apart from existing studies such as McSweeney & Jones (2016) and how your results fit into the context of the existing literature (#2, #14).
Also, we have added a section to discuss the results and provide our considerations.
Please find below.

**--- Specific comments**
☐ Section 1 Introduction
1. P2 L. 19: Please specify what these previous studies have found, and why we have yet to reach a consensus on the method to select small ensembles. Also, revisiting these papers in a discussion section would provide the necessary context to interpret the results and whether the methodology used in this study is distinct from, or improves upon, the ones used in these studies.
In the previous study, the condition for selecting subsets depends on their purpose. For example, whether the model performance is considered, which climatological or extreme variables are used and which region is interested. Thus, we have yet to reach a consensus. Our purpose in this study, however, is to

indicate the property of ISIMIP and CORDEX subsets for the ability to reproduce the present-day temperature and precipitation and for their future change, and is not suggestions of model selection methodology (P3 L23-25). In P2 L19-21, we have described that, although there are various methods, it is most desirable for the methods to select subsets of GCMs that have smaller biases in the historical climate simulations and cover the widest possible uncertainty range of future projections. We have discussed whether the current subsets in ISIMIP and CORDEX are such a subset.

We have just modified the related sentence as the response to "why we have yet to reach a consensus on the method to select small ensembles";

(P2, L19) "... Gobiet 2016). The optimum method, however, remains to be determined because the interests depend on the studies, for instance, how the model performance is considered, which climatological or extreme variables are used and which region is interested."

2. P3 L. 23-36 Please clarify what is specific to this study: is some aspect of the methodology new? Is this performing an existing analysis to a new set of data? Is the added value of the manuscript to specifically address whether region-specific subsets (CORDEX) outperform a globally consistent sub-set (ISIMIP)? Stating this explicitly would improve the value and readability of the manuscript.

The methodology is not new. The analysed subset has been changed from the subset analysed in McSweeney and Jones (2016) by following the updated selection in ISIMIP. The added value of our manuscript is following, which have been added to P4 L2:

(P4 L2) "The ability for the ISIMIP subset was not mentioned by MJ2016 and thus we investigated that in region by region. We analysed four GCMs selected in ISIMIP2b (unless specified otherwise, hereafter refers to as ISIMIP) here. Thus, assessment of the projections was also updated from MJ2016. The GCMs used in CORDEX have been assessed by region in previous studies, but are limited (e.g., Haensler et al. 2013 for Africa; Bartók et al. 2017 for Europe; Karmalkar 2018 for North America). Even simple assessment conducted is needed for the present CORDEX. Furthermore, uniform assessment across regions permits to discuss the difference of characteristics among the regions and the possibility of heterogeneous scenario as mentioned above. By using the subsets from the two programs, we can explore the difference between the original subset in CORDEX and the subset selected with assuming CORDEX CORE, which is helpful information for the model selection in CORDEX CORE."

From an additional analysis in this revision, we suggest that nine models are needed to solve the heterogeneity of the uncertainty. This result can provide suggestions to the next generations of model selections.

(P9 L18) "The current CORDEX subsets can capture both uncertainties for temperature and precipitation in the regions with a relatively large ensemble. However, it is found that changing the number of models from the current CORDEX members to nine members can capture more than half of the full uncertainty in both projections of temperature and precipitation in more than 85% of all regions, with a possibility of 50%. Furthermore, the same is also shown as for the ISIMIP subset, but for 70% of all regions. Focusing the uncertainty in the future projections, this result proposes that the current number of models need to be changed to discuss a similar uncertainty range among the regions."

□ Section 2.1

3. P4 L. 18 Why focus on land area only? Regional precipitation, including over the seas/ocean, is relevant for impact studies. Please clearly state the scope (and the application) of this study.

The impacts of climate change appear over the land and ocean as you mentioned. The reason why focusing on land is that the assessment sectors in ISIMIP are mainly over land, and it is important for both programs because of the relevance to human activities. We have added the sentence below:

(P4, L18) "… we focused on the global land area, considering the importance for both programs because of the relevance to human activities."

4. P4 L. 22 Can you justify why you excluded low-precipitation models from the precipitation analysis? I understand these models significantly bias your ensemble but this undermines the stated

aim of the study (i.e. to quantify the spread of the full CMIP5 ensemble covered by the ISIMIP and CORDEX subsets). You arbitrarily reduced the model spread covered in this study. Please at least provide more information as to how many models were excluded (and thus the size of your remaining ensemble), why 0.1mm/day was chosen as a threshold, and the reasoning to exclude these models from the precipitation study but to keep them for temperature (if the argument is that the climate they produce is too unrealistic to be a plausible representation of today's climate).

We have expressed the future change of precipitation as a change ratio of the future precipitation to the present-day precipitation. The expression, which has been often used, is highly sensitive in dry grids. Even if the change amount is quantitatively small, the ratio is extremely large. Such a large ratio leads to a large regional average. The large ratio by a small change in dry grids is difficult to explain the validity, and thus we took the dry grids out of consideration. The threshold can be permitted the exclusion of grids with the ratio of 100% over around the Sahara, and be suppressed the exclusion under 5% of all analyzed grids. We added the following sentence to explain the exclusion:

(P4, L22) "The future change of precipitation expressed in a ratio here. That is the change ratio tend to be large at too dry grid even when the change is quantitatively extreme small. Such a large ratio is difficult to explain its meanings physically. By applying the threshold, the grid indicating an extremely large ratio, for instance, 100% were excluded. The total number of the excluded grids is approximately 5% of all target grids as an average over the used members."

5. P4 L. 32 Please include a definition of the skill score used here, or a reference to a published paper using the exact same skill score. In Taylor (2001), two examples of skill scores are used, to illustrate that the skill score can be adjusted depending on whether you value high correlation or matching the variability most. In addition, it is explicitly stated that the value of Ro should be reported every time a skill score is used.

We appreciate your pointing out. The definition below has been added to P4, L32,

(P4, L32) "… we used the skill score proposed by Taylor (2001) (hereafter referred to as skill score) as follows:
$S=4(1+R)/\{(\sigma+\sigma^{-1})^2(1+R_0)\}$, (1)
where R is the spatial correlation coefficient between referred observation and simulation, $\sigma$ is the standard deviation of simulation normalized by the reference spatial pattern and $R_0$ is the maximum correlation attainable. The value of $R_0$ was assumed to 1 here."

6. P5 L5 Is R the min-max range of a given subset? Please clarify. Using 'uncertainty range' is misleading; it sounds like you are sampling your ensemble. If I understand correctly, you generate 10,000 values of RSub, then look at ensemble spread (in Fig 4).

The value, Rsub which we used here, is the max.-min. ranges of the uncertainty estimated from the ISIMIP subset, the CORDEX subset, or the 10,000 random subset samples from $CMIP_{Full\_Future}$. The corresponding parts have been modified as follows:

(P5 L4) "The FRC from the regional averages (FRA) was defined as the fraction of the maximum-minimum range of the uncertainty in the regional averaged projections from a subset of $CMIP_{Full\_Future}$ ($R_{Sub}$) to the range from $CMIP_{Full\_Future}$ ($R_{Full}$), as follows:
(Equation 2)
The range of $R_{Sub}$ was computed from the ISIMIP and CORDEX subsets and also arbitrary subset samples we generated. From the comparison with the arbitrary samples, we can investigate how well the ISIMIP and CORDEX subsets captured the uncertainty range of projections. McSweeney and Jones (2016) presented the comparison using their 500 samples as 'representation'. Our arbitrary samples were generated by randomly selected n models without repetition from $CMIP_{Full\_Future}$ 10,000 times, where n is the sample size of subsets in ISIMIP (n = 4) or CORDEX (n depends on the regions; see Table 1). Then, the variance of the FRA was estimated from the 10,000 random samples of the subset of $CMIP_{Full\_Future}$ and compared with the FRA from the ISIMIP and CORDEX subsets."

☐ Section 3

7. This section is entitled 'Results and Discussion' but mostly contains the description of the results. Regardless of whether it is included in this section or a separate section, the manuscript needs a more comprehensive discussion (see other comments below).

We have made an additional section in this revision for the discussions. Please check the responses below.

☐ Section 3.1

8. P.5, L. 28-29 Please include the top 50% ensembles in the supplementary material, so that future model selection can rely on your analysis to select less biased models.

We appreciate your constructive suggestion. We have added the high performance subsets as an supplementary material. The material has been refereed in P5, L19 ("The models included in the high performance subset is shown in Supplement 3.").

10. P5 L29-30 Can you suggest why the spread is different between the two ensembles over the Northern Hemisphere? In Fig.1, the spread for ISIMIP is sometimes significantly larger, sometimes significantly smaller than CORDEX. Could this be due to the number of models in CORDEX? Do some of the regions have models that overlap across ISIMIP and CORDEX? Simply stating that they are different in some regions is not very informative.

We appreciate your accurate indication. As you pointed out, we have found that part of the characteristics of the difference in the spread has a relationship to the overlapping of used model members. The sentence has been modified and added more explanation:

(P5 L29-30) "The difference in the spread between the ISIMIP and CORDEX subsets has a characteristic in region-by-region and part of them relates to the overlapping of model members used across ISIMIP and CORDEX. For example, in five regions of Central and South America, Europe, Africa and South Asia, the CORDEX subsets include more than three of four ISIMIP models and the ensemble is large in CORDEX than in ISIMIP (Supplement 1). As the result, the variance of biases estimated from the CORDEX subset covers that from ISIMIP. Especially in Europe, the difference of the variance between the CORDEX and ISIMIP subsets is large and the models not included in the ISIMIP subset are found to make the variance increase. Focusing on the regions where the CORDEX subsets include only two models in the ISIMIP subset, the variance from the CORDEX subset tends to be larger than that from the ISIMIP subset, especially in the regions with large ensemble in the CORDEX subsets, like North America, SEA and Australasia. By contrast, the variance from the CORDEX subsets is relatively small in the regions with small ensemble in the CORDEX subsets, like MENA and Central Asia. In East Asia, the variance is small in CORDEX despite using seven models in contrast to four models in ISIMIP. Thus the biases are found to be similar to each other in CORDEX-East Asia."

Also, we have modified the sentence about the spread of the temperature bias:

(P6 L9-11) "The spread of B(T(ISIMIP)) is covered by that of B(T(CORDEX)) in the same four regions as the bias in the precipitation except for Europe, because of the overlapping of used model members. The spreads of B(T(ISIMIP)) and B(T(CORDEX)), however, resemble each other compared with the precipitation bias, indicating that CORDEX used models with a quantitatively similar performance to ISIMIP, despite using more models than ISIMIP except for Central Asia."

11. P6 L1-3 This paragraph would benefit from an earlier explanation about how the skill and bias metrics are different and the insight gained by using both. Include some interpretation of why the model ensembles that perform relatively well in a bias metric perform less well in the skill metric. (same comment for P6 L11)

Thank you for your suggestion. The skill score quantifies the similarity of the spatial pattern by a correlation coefficient and a standard deviation. The bias evaluates the quantity itself by the regional average of the difference from the observation. Thus there is a case with large positive and negative biases in each grid even when the spatial average is small, that is to say, the spatial pattern is different from the observation. The ensemble showing a small bias and a low score represents the quantity closed to the observation as the spatial average but a low similarity of the pattern. Therefore both metrics are

needed to assess how well the ensemble represents the reality. We have added the following sentences in each part:

(P4, L32) "In addition to the skill score, we use the model bias to evaluate the quantity itself. The usage of the two metrics enables the assessment of both the spatial pattern and the quantity."
(P6, L2) "That is to say, ISIMIP and CORDEX subsets include the member showing a low similarity of the spatial pattern to the observation."
(P6, L11) "Therefore, relative to $CMIP_{highS}$, the subsets can quantitatively represent the observed temperature as a regional average well but the spatial pattern represented by some members in the subsets has not much resembled the observation."

12. P6, L. 15-16 Please include the top 50% ensembles in the supplementary material. In addition, please include in the discussion whether the 'best performing models' perform well both in temperature and precipitation, and whether selecting according to high skill or low bias makes a difference. As you state that a better ensemble can be selected, please give the evidence from your results that this can be done robustly.

We appreciate your constructive suggestion. We have added the top 50% models as Supplement 3 (Response #8). The comparison between the top 50% ensembles for the bias and skill score is interesting. From Supplement 3, when we focus on one variable of either temperature or precipitation, 13 models in 25 all high-performance models are included in both subsets of high-performance models for the bias and skill score. Thus, the model with a small bias indicates a high score with 50% of the possibility. We have described this explanation to Section of discussion which we have added in this revision:

(P8 L23) "Focusing on one variable of either temperature or precipitation, 13 models in 25 all high-performance models are included in both subsets of high-performance models for the bias and skill score (Supplement 3). In addition to the two indices of bias and skill score for one variable, the models indicating the high performance for both two variables of temperature and precipitation is 0 at the minimum number in Southeast Asia and the Arctic and 9 at the maximum number in Africa. The averaged number over the regions is approximately 4. Therefore, although the model with a small bias indicates a high score with 50% of the possibility, it is difficult to select models with a high performance for both variables of temperature and precipitation."

In addition, explanation and discussion were not enough for the description of selecting a better ensemble. We have added the limitation.

(P8 L23) " … a much better model subset, regarding to biases and skill scores, can be selected with making use of the advantage of the small number of models. However, such a selection can be conducted when there are no constraints of data availability which was the main constraint to select the current subsets in ISIMIP and CORDEX and when we use one variable of either temperature or precipitation."

☐ Section 3.2
13. P6, L.25 Please place this into context by mentioning other studies that have looked at emergent constraints, even if it's only in specific regions (e.g. Bracegirdle et al, 2018 for Southern Ocean winds; Bracegirdle and Stephenson 2013 for Arctic warming).

As you noted, we have added related previous studies and modified the sentence,

(P6 L25) "… suggesting that the bias and skill score are not good emergent constraints to reduce the uncertainty of $\varDelta T$ in this study though the previous studies have showed the reduction of the uncertainty (e.g. Smith and Chandler 2010; Bracegirdle and Stephenson 2013; Bracegirdle et al., 2013; Simpson et al. 2016)"

☐ Section 3.3
14. In general, this section is confusing. It would benefit from clearly stating what is being compared, and referring to specific aspects of Fig 4 to support your statements. Specifically: P8 L1-2: Please clarify which metric you use to make that statement (i.e. the total coverage on the y-axis). I got

confused because the performance of FRACORDEX remains low compared to FRARandom_C, even as the number of models increases. Please state explicitly where you are comparing it to the full range, or to Random_C (3 sentences later). P8 L16: Please specify which FRA you are talking about: the median of FRARandom_C? Or FRACORDEX? Or both? P8 L19-22: This is an interesting point, but if you make the point that increasing the number of models produces a higher FRA, please show the evidence for it. The latter part of the description is unclear so adding the technical details and a figure would make a stronger point.

We have modified each sentence you pointed out as what follows:

(P8 L1-2) "A relatively high coverage, above ~50%, is shown on $FRA_{CORDEX}$ for both changes of temperature and precipitation in eight regions when using nine models or more, except for temperature in Antarctica (Fig. 4a, b): that is to say, the CORDEX subset captures more than half of the range from $CMIP_{Full\_Future}$."
(P8 L16) "…and thus the large model ensemble results in an increase in $FRA_{CORDEX}$ and $FRA_{Random\_C}$."
(P8 L19-22) We have added the figure for the change of FRA with the number of models not only in Central Asia but also in the other regions. Please check Figures 5 and 6.

☐ Summary and Conclusions
15. This section provides a general summary of the findings, but would benefit from providing context as to how these results compared to other studies (e.g. those cited in the introduction), and how the findings advance the general understanding of the field. In addition, statements in P9 L4-5 and P9 L15-16 seems to indicate CORDEX performance to be bad relative to randomly selected ensembles, while P9 L8-9 states 'relatively wide coverage of both uncertainties'. Please clarify so that the message is not ambiguous. For example, it is ok to state that CORDEX is not performing well compared to the randomly selected ensembles, but is marginally better than ISIMIP at sampling uncertainties in projected change in precipitation.

Regarding the comparison with other studies, as mentioned in P7 L9, "... global consistent four models used in ISIMIP2b, which are taken into consideration of the ability of reproduction, still remains difficult to capture the uncertainties in regional precipitation change, as in McSweeney and Jones (2016) which analysed for five models in the fast track." The result of assessing the CORDEX subset was not able to compare with other studies because of the difference in the variables, part of the regions and seasons. For results from an additional analysis conducted in this revision, we referred to the approach in McSweeney and Jones (2016) but the results couldn't compare each other. It is because "They focused on a subset covering the uncertainty in each grid most widely over the globe or regions and investigated how the coverage changes with the number of models. On the other hand, in this study, to consider making better use of the current subsets, we investigated how the coverage changes with changing the number of models from the current model members." (Added to Section 3.3)

Thanks for your suggestion on the ambiguous statement. We have added the following statement to P9 L14 and have modified the statement on P9 L15-16.
(P9 L14) "The CORDEX subset is not performing well compared to the randomly selected samples but is marginally better than ISIMIP at covering uncertainties in the projected change in precipitation when a large model ensemble used."
(P9 L15-16) "The region-specific model subset, like CORDEX, captures coverage of both uncertainties well compared to the global common subset, but large model members are needed."

16. Please include a more comprehensive discussion of your methods and results, including:
    Two metrics for "good performance" are used in parallel throughout the study (low bias and high skill score). Please comment as to how similar/distinct these two metrics are, and on the insights gained by using both (qualitatively or quantitatively). Similarly, how different are the 'top 50%' ensembles? i.e. does using skill or bias for selection of the best performing models significantly affect the ensemble?
We appreciate your comments.
First, how similar/distinct these two metrics are, and on the insights gained by using both?

As described in Response #11, we can evaluate the abilities to represent the spatial pattern and the quantity itself by using the two metrics. How similar these two metrics are can be estimated by how many the models selected by each metric overlap. Because 13 models in 25 all high-performance models are included in both subsets of high-performance models for the bias and skill score, the similarity is not so high, around 50%. The number of overlapped models is described in Section of discussion added in this revision. (P8 L23)

Second, how different are the 'top 50%' ensembles?
How different are the top 50% ensembles have been shown in Response #11. Please check. The model with a small bias indicates a high score with 50% of the possibility. Thus a significant influence appears on the selected ensembles.

17. The results section 3.1 focuses mostly on whether the ISIMIP and CORDEX fall within the observational spread (e.g. L. 2-3 on page 6). It would be helpful to distinguish whether this is mainly due to a large spread in the model ensembles, or whether a systematic bias is seen in certain regions (e.g. Fig 1 shows model ensembles overestimate precipitation in most regions). Also, please include a discussion of the expected variance of model ensembles. In coupled models, the timing of climate variability modes is unlikely to match that of observations, so the variance over a 20-year period is likely to be higher in model ensembles than observations.

Here, the observational spread is the spread of the 20-year averaged precipitation calculated from seven observational datasets, not the variance over a specific period. Therefore we cannot discuss the different variance between the model and observations, resulted from the timing of climate variability.

18. In this study, the performance of CORDEX and ISIMIP are considered independently for the temperature and precipitation changes (with precipitation being scaled by the temperature change). Please discuss whether there is any evidence that a selection on one variable (e.g. precipitation) is sufficient to select good performing models, or whether a combined approach is necessary to select models. In climate impacts, people care about the plausibility and diversity of climate sampled, not a single variable.

We appreciate your important suggestion. In this revision, we confirmed a quite small number of models indicating a high performance for both principal variables of temperature and precipitation. In addition, we considered that the evaluation for the simulated principal variables is needed for the studies of ISIMIP and CORDEX, but not possibly sufficient for model selections. Because the large-scale circulation characterized the regional climate, its performance is also important. When we can obtain the reference data, the method used in this study can be applied to the evaluation of the performance. To select subsets in the next generations with the performance considered, it is necessary to construct a combined approach that can take into account multiple variables. We have described this explanation to Section of discussion which we have added:

(P8 L23) "…Therefore, although the model with a small bias indicates a high score with 50% of the possibility, it is difficult to select models with a high performance at the quantity and the spatial pattern for both variables of temperature and precipitation.
In this study, we assessed the current ISIMIP and CORDEX subsets to investigate whether the subset indicates small biases in the historical climatology and covers the uncertainty in the future projections widely using temperature and precipitation. Both variables are most frequently used in future projections and also weather forecasts. The evaluation for such a principal variable is important for the studies of ISIMIP and CORDEX. It should be noted, however, that ISIMIP needs the dataset with reasonable for multiple variables used in their impact assessment and with enable to discuss the uncertainty in the projections. CORDEX requires the dataset with based on a plausible mechanism of the climatology as the input data for RCMs. Thus, there is a possibility that a good subset which we presented based on the model performance for temperature and precipitation would be an option of their future subsets.
Although ISIMIP and CORDEX have tight constraints for model selection at the present, both programs will select the subset showing a reasonable climate based on a plausible mechanism in the future. In the case, two variables of temperature and precipitation are not possibly sufficient for model selections. At least for the regional climatological studies and the assessment of its impact, it is important to reproduce

large-scale circulations which characterize the regional climate. Especially, the spatial pattern of precipitation depends on the accuracy of the circulation. Indeed, model change in ISIMIP from the fast track to ISIMIP2b has already been performed with a consideration of the ability to reproduce ENSO and monsoon (Frieler et al. 2017). The evaluation method used in this study can be applied to the other variables when we can obtain the reference data. For instance, Taylor's skill score which we used to evaluate the pattern of temperature and precipitation can also apply to the pattern of circulation. However, as more variables and evaluation indices are employed, it is more difficult to obtain the CMIP5 models with high accuracy as described above.

It is preferable to select subsets in the next generations based on a combined approach that can consider not only the ability to reproduce the principal variables of temperature and precipitation but also the other ones which are also important to characterize the regional climate. Construction of such an approach would be one of the important tasks for both programs."

In addition, we described the following sentence to Summary:

(P9 L18) "In this study, we have assessed the subsets using the principal variables of temperature and precipitation. It is not sufficient for selecting subsets in the next generations. We suggest that it is preferable a combined approach that can consider the ability not only for temperature and precipitation but also for the other ones which are also important to characterize the regional climate. Construction of such an approach would be urgently demanded for both programs."

**--- Technical corrections**
P1 L18 (and after) High performed models -> high performance models or high-fidelity models
We have modified them as the referee mentioned. Thanks.

P1 L20-25 Please rework this section to clarify the meaning. As you have not previously introduced the 10,000 sampling strategy, these two sentences are confusing.
I am sorry for the confusing. The section has been rephrased,
"Compared with the randomly selected 10,000 arbitrary subset samples, the CORDEX subset shows low coverage of the uncertainty for the temperature change projections in some regions, and the ISIMIP subset high coverage in all regions. On the other hand, for the precipitation change projections, the CORDEX subsets show low coverage in half of the regions compared with the arbitrary subsets, but tend to cover the uncertainty widely compared with the ISIMIP subset."

P2 L33 Please rephrase this sentence for better readability. For example: "In addition, paper X and Y showed that combining region-specific subsets covers more uncertainty than a single, globally consistent, subset of models."
The sentence has been rephrased as follows:
"They also illuminated that region-specific subsets generally cover more the uncertainty than globally consistent subsets in 26 global regions."

P5, L11 Please rephrase that last sentence for readability.
The sentence has been rephrased as follows:
"Then, the variance of the FRA was estimated from the 10,000 random subset samples of $CMIP_{Full\_Future}$ and compared with the FRA from the ISIMIP and CORDEX subsets."

P7, L9-12 This sentence needs to be reworked for readability.
The sentence has been rephrased as follows:
"Therefore, the subset of four models used in ISIMIP2b shows the difficulty of capturing the uncertainties in regional precipitation change. This result is the same as stated using the subset of five models used in the fast track of ISIMIP discussed by MJ2016, despite two of the five models changed."

P7, L15 "with those of the 10,000" -> remove "the"
We have modified them as the referee mentioned. Thanks.

P7 L16 "randomly sampled subsets" of what?
We have rephrased to "randomly samples subsets of CMIP$_{Full\_Future}$".

P9 L17 Be more specific: 'it depends on the number of models used' is too vague to be informative -> "FRA increases with the number of models used", or "regions covered by bigger ensembles generally have higher FRA". . .
We agree with the comments. The sentence has been modified to "large model menber are needed".

P13 L8 "areal mean of the reference data" -> normalized by the regional average of GPCC data.
We appreciate your revised. We have modified the sentence as you mentioned.

P14 Figure 2 Why does Antarctica have no top 50% in temperature? Explain that somewhere (main text or figure caption).
We apologize for missing the explanation. The reference data of temperature does not cover the Antarctica, and thus we cannot indicate the results for the top 50%. We have added the sentence below in the caption of Figure 2 and also Supplement 4.
"The top 50% of the CMIP5 models cannot be plotted over Antarctica because of missing the CRU reference data."

P16 Figure 4 "uncertainty range" -> range Also, why are red dots missing in some regions in Fig 4a?
We have changed to "range". Red dots look missing because the dots overlap where the coverage is the same between ISIMIP (blue dot) and CORDEX (red dot). "The ISIMIP and CORDEX coverages in (a) overlaps in MENA, N. America and Africa." is added to the caption.
* * *
We have revised our manuscript to address comments from Anonymous Reviewer #2.

---

## Referee Report (RR1)

Referee Report on "Uncertainties in climate change projections covered by the ISIMIP and CORDEX model subsets from CMIP5" by Ito et al.

**--- General comments**

This manuscript presents a valuable study quantifying the spread of CMIP5 projections and biases covered by the subsets of models used in the ISIMIP and CORDEX experiments. This revised manuscript does a better job at explaining the added value of this study compared to previous studies.

Several points are raised in the body of the manuscript, which would benefit from a mention in the discussion section. These include: (i) the evaluation and selection of the high performance models was done independently for the two metrics used, i.e. temperature and precipitation. A next step would be to evaluate T and P jointly. Can the authors suggest whether that would impact the model selection? (ii) Similarly, the randomly selected ensembles can perform well in one variable, but could be evaluated on both variables. E.g. could one set of model perform well for both variables, including $\Delta T$ and $\Delta P$? Adding a few sentences to mention (i) and (ii) as limitations of the study and/or perspectives for future research would strengthen the discussion. (iii) The authors mention using "other variables" in addition to T and P, but do not specify which ones.

This manuscript would benefit from adjustments to the language and sentence structure, to improve the readability of the final manuscript. Most of the minor comments below aim to remove ambiguity and improve the flow of the manuscript.

**-- Specific comments**

P1 L17-18, "However, the spreads in…"
P1 L21, "with the randomly selected 10,000 arbitrary subset samples" → with 10,000 randomly selected subset samples
P3 L2-4 " the subset covers… randomly sampled" → the subset covers more of the uncertainty in the temperature and precipitation changes projected by 36 CMIP5 GCMs, than other randomly sampled five-GCM subsets.
P3 L30, "how extent" → to what extent
P4 L3-11, this paragraph is difficult to follow due to language and sentence structure. Please rework this paragraph to improve readability.
P4 L31 what do you mean by 'too dry' grids? It is unclear whether grid cells or entire models are excluded. If it is entire models (with spatially averaged precipitation < 0.1 mm/day), these models should still be included in the study of the historical bias, and excluding them should only be applied to the study of the projected changes. Currently, it is unclear whether these models are excluded entirely.
P4 L32, then → the ; "mean precipitation" → specify which mean (spatial mean, spatial and time mean?)

P5 L1 and L3, "grid" is ambiguous, please replace with "grid cells" or "models" depending on what you mean.

P5 L7, "multi-precipitation products"; it is unclear here whether you combine all these products (e.g. by averaging them) into a single reference product. According to the caption of Figure 1, all 6 were used in the "obs" ensemble, but GPCC is your reference product. I suggest amending "multi-precipitation products" → " six different precipitation products", and adding a line saying that all the precipitation  biases were calculated using GPCC as the reference.

P5 L17, referred → reference

P5 L19, "we use the model bias to evaluate the quantity itself" →  we also evaluate the magnitude of the model bias. Using both metrics enables… the spatial pattern and the bias magnitude.

P5 L28, also arbitrary → also from arbitrary…

P5 L30, "MJ2016 presented...representation." This sentence is awkward and does not add anything. Remove or adjust → "MJ2016 presented a similar comparison between the original ISIMIP model data and 500 randomly selected five-model ensembles".

P6 L9, "biases: the other" → "biases; the other"

P6 L24, large → larger. "As the result" → As a result

P6 L32, "four models in ISMIP. Thus…" → four models in ISMIP, indicating that the biases… are almost the same.

P7 L3, "include... observation" → include members showing a spatial pattern of low similarity to that of observations.

P7 L7 "would be related" → could be related

P7 L16 " has not much resembled the observation" → does not resemble that of the observations.

P7 L25 "change of " → changes in

P7 L 26 "projected increments of the temperature". Ambiguous. I suggest →  average rate of temperature increase per year, calculated from the 20-year period for each model

P7 L30, Any suggestion of why this result is different from other studies? It would be good to include some mention of this in the discussion section.

P8 L2 "with totally covering" → and cover the full range…

P8 L5 "like at least the bias.." → for example whether we use the bias or the skill score.

P8 L7, "capture the full range less... " → capture less than 60% of the full range in all regions.

P8 L8, capture the wider → capture a wider range

P8 L8, "subsets, differing from…" → subsets, a result markedly different than for $\Delta T$, where both CORDEX and ISIMIP have relatively large coverage.

P8 L9, shows the difficulty of → has difficulty capturing

P8 L12, → uncertainty range (maximum-minimum) is …

P8 L16, Only in… → In Central Asia, the full range of.. remains below the 25th percentile… while the maximum-minimum range of .. adequately covers the IQR… Thus, the three models.. Central Asia underestimate the average tendency…

P8 L19 "despite being… differing from ISIMIP" This is unclear. Please rephrase.

P8 L24 "high coverage for the temperature.." How about precipitation? Please include the precipitation results for ISIMIP too.

P8 L 31 presents distribution → represents the distribution…

P9 L32, "They focused… models" Unclear, please rephrase. In particular, "grid" is ambiguous, I suggest using model (or grid cell) if applicable.

P10 L8, "which are less than nine members" this fragment does not add anything. Please remove or adjust to explain why nine members is relevant.

P10 L24, "high performed models" →  high performance models.

P10 L25, "regarding to" → with regard to

P10 L29, "13 models in 25 all high performance models" → 13 high performance model (out of 25) are included…

P10 L32 "Therefore, although… possibility. " Unclear, please rephrase.

P11 L3, the subset → their model ensemble indicates…

P11 L7, the dataset with reasonable for… projections → a dataset with reasonable values for… and with enough coverage of the projection uncertainties.

P11 L8 with based → with values based

P11 L12, Remove "in the case". Also, can you suggest other relevant variables, given that you state temperature and precipitation are not enough?

P11 L19, similar to the above, can you include a suggestion of a variable to characterize "circulation" (e.g. geopotential height, sea level pressure…)?

P11 L23, similar to the above, simply stating "the other ones" is too vague. Make specific suggestions of potentially useful variables.

P12 L3, How about using both precipitation and temperature? Can you comment here on a combined evaluation and whether the results are likely to be different from the ones included in this study?

P12 L20, "with capturing" → while capturing

P12 L22, "the global common subset" → the global common (ISIMIP) subset ; "but large model ensemble is needed" → but performs better when a large number of models is used.

P12 L26, "need to be changed" to what? Specify, e.g. increased to seven, or nine models if possible….

P12 L31, specify which other variable might be relevant.

Figure 4: The red dot is missing for Central Asia. Is it also an overlap? If so, change the caption accordingly. For the 3 regions where the dots overlap, I suggest changing the size of one of the dots (e.g. the red dot slightly larger than the blue dot), so that the outside of the red dot is visible, this would make the plot easier to understand by showing the overlap, rather than having to read the caption.

---

## Author Response (AR2)

Response to the comments from Editor for the manuscript:
"Uncertainties in climate change projections covered by the ISIMIP and CORDEX model subsets from CMIP5" by Ito et al.

We would like to appreciate your careful review and constructive comments that helped to improve our manuscript. We have revised our manuscript in accordance with your comments. Please check our detailed responses below. Editor comments below are shown in black and our responses are in blue. Page and line numbers below are corresponding to the number in the revised manuscript.

--- General comments
This manuscript presents a valuable study quantifying the spread of CMIP5 projections and biases covered by the subsets of models used in the ISIMIP and CORDEX experiments. This revised manuscript does a better job at explaining the added value of this study compared to previous studies.

Several points are raised in the body of the manuscript, which would benefit from a mention in the discussion section. These include: (i) the evaluation and selection of the high performance models was done independently for the two metrics used, i.e. temperature and precipitation. A next step would be to evaluate T and P jointly. Can the authors suggest whether that would impact the model selection? (ii) Similarly, the randomly selected ensembles can perform well in one variable, but could be evaluated on both variables. E.g. could one set of model perform well for both variables, including ΔT and ΔP? Adding a few sentences to mention (i) and (ii) as limitations of the study and/or perspectives for future research would strengthen the discussion. (iii) The authors mention using "other variables" in addition to T and P, but do not specify which ones.
We appreciate your positive evaluation and useful comments. We have made our point-by-point responses to the comments. The head number is corresponding to three comments.

(i)   From the discussion using the supplement 3, "the number of models indicating the high performance for both two variables of temperature and precipitation is 0 at the minimum in Southeast Asia and the Arctic. (P11 L5)" Thus, "it is difficult to select such models for both variables of temperature and precipitation. (P11 L7)" Temperature and precipitation are a principal variable for studies of ISIMIP and CORDEX (P11 L11). However, especially the quantity and the pattern of precipitation is given by elaborate model schemes and accurate representations of the circulation patterns and thus, representing the realistic is still difficult. Therefore, it might be good that the evaluation of sea level pressure indicating the circulation, instead of precipitation, as the first step of the combined evaluation though there remains a missing of the reference data. We have added this point to P11 L28,
"As the first step of the combined approach, it could be good that the evaluation of sea level pressure indicating the large-scale circulation which has an influence on the precipitation pattern, instead of precipitation itself. This combination might obtain an adequate number of members, which is found to be difficult using the combination of temperature and precipitation here."

(ii)   Though this comment is tough to respond because we did not do such an analysis, Figure 4 presents that the coverage of $FRA_{Random\_C}$ is relatively high on both variables when the number of members is large. That is, it is found that the possibility to cover the projection uncertainties for both variables gets high by applying a region-specific ensemble and an adequate number of its ensemble members. We have added this point to P11 L31,
"Regarding combined approaches for future changes, Figure 4 presents that the coverage of $FRA_{Random\_C}$ is relatively high on both variables when the number of members is large. Thus, there would be the possibility to cover the projection uncertainties for both variables widely by applying a region-specific ensemble and an adequate number of its ensemble members."

(iii)   We indicated the importance of consideration of large-scale circulation patterns but did not specify the variables. Two variables of sea level pressure and geopotential height for expressing the patterns have been added to the discussion (P11 L25 and P11 L29) and our conclusions (P13 L12). This

modification is also corresponding to the response to your specific comments below.

This manuscript would benefit from adjustments to the language and sentence structure, to improve the readability of the final manuscript. Most of the minor comments below aim to remove ambiguity and improve the flow of the manuscript.
We appreciate your comments. Below are our responses to each of your points. According to the comments, we have modified our manuscript.

-- Specific comments
P1 L17-18, "However, the spreads in..."
We have modified to,
"However, the ISIMIP and CORDEX subsets show the larger spread than high-performance models from the full set, despite using a small number of models in ISIMIP and CORDEX." (P1 L17)

P1 L21, "with the randomly selected 10,000 arbitrary subset samples" → with 10,000 randomly selected subset samples
We have done (P1 L21). According to this modification, P1 L24 also have been changed, "the arbitrary subsets" → "the randomly selected subsets".

P3 L2-4 " the subset covers... randomly sampled" → the subset covers more of the uncertainty in the temperature and precipitation changes projected by 36 CMIP5 GCMs, than other randomly sampled five-GCM subsets.
We have modified. (P3 L2)

P3 L30, "how extent" → to what extent
We have done. (P3 L30)

P4 L3-11, this paragraph is difficult to follow due to language and sentence structure. Please rework this paragraph to improve readability.
Thank you for pointing out. We have modified the paragraph to the below.
"Regarding the ISIMIP subset, there are two updated points from MJ2016. One is the investigations of the ability to represent historical climate for the ISIMIP subset, which MJ2016 did not mentioned; another is that our target GCMs are four GCMs selected in ISIMIP2b (unless specified otherwise, hereafter refers to as ISIMIP). Regarding the CORDEX subset, previous studies have assessed the GCM simulations in some regions, but are limited (e.g., Haensler et al., 2013 for Africa; Bartók et al., 2017 for Europe; Karmalkar, 2018 for North America). Therefore, even a simple assessment of GCM simulations is needed for understanding their downscaled simulations.
Uniform assessment across regions enables to discuss the regional characteristics and the possibility of heterogeneous scenario among regions as mentioned above. Furthermore, by using both subsets from ISIMIP and CORDEX, we can explore the difference between the original subset in CORDEX and the subset assuming CORDEX CORE (global common subset), which could be helpful information for the model selection in CORDEX CORE." (P4 L3)

P4 L31 what do you mean by 'too dry' grids? It is unclear whether grid cells or entire models are excluded. If it is entire models (with spatially averaged precipitation < 0.1 mm/day), these models should still be included in the study of the historical bias, and excluding them should only be applied to the study of the projected changes. Currently, it is unclear whether these models are excluded entirely.
We have defined the grids < 0.1 mm/day as too dry grids, and the precipitation value at the grids were out of the consideration. Thus, there are no models which are excluded entirely. We have changed the sentence P4 L32-P5 L3,
"The grid cells with the temporal mean precipitation of < 0.1 mm/day were defined as 'too dry' grid cells and the precipitation value at the cells was out of consideration. It is because we expressed the precipitation change in a ratio and thus the ratio tends to be large at too dry cells even when the change

is quantitatively extreme small.”

P4 L32, then → the ; “mean precipitation” → specify which mean (spatial mean, spatial and time mean?)
We have changed from “the mean precipitation” to “the temporal mean precipitation.” (P5 L1)

P5 L1 and L3, “grid” is ambiguous, please replace with “grid cells” or “models” depending on what you mean.
We have changed to “grid cells.” (P5 L1)

P5 L7, “multi-precipitation products”; it is unclear here whether you combine all these products (e.g. by averaging them) into a single reference product. According to the caption of Figure 1, all 6 were used in the “obs” ensemble, but GPCC is your reference product. I suggest amending “multi-precipitation products” → “ six different precipitation products”, and adding a line saying that all the precipitation biases were calculated using GPCC as the reference.
The sentence has been changed to “seven different precipitation products” and the following has been added to P5 L15,
“The difference among the observations was calculated as the deviation from GPCC as the reference.”

P5 L17, referred → reference
We have done. (P5 L19)

P5 L19, “we use the model bias to evaluate the quantity itself” → we also evaluate the magnitude of the model bias. Using both metrics enables... the spatial pattern and the bias magnitude.
We have done. (P5 L 21)

P5 L28, also arbitrary → also from arbitrary...
We have done. (P5 L30)

P5 L30, “MJ2016 presented...representation.” This sentence is awkward and does not add anything. Remove or adjust → “MJ2016 presented a similar comparison between the original ISIMIP model data and 500 randomly selected five-model ensembles”.
Thank you for your comments. We removed the sentence and the sentence, “With reference to MJ2016, ” have been used. (P6 L1)

P6 L9, “biases: the other” → “biases; the other”
We have done. Thank you for your careful reviews. (P6 L10)

P6 L24, large → larger. “As the result” → As a result
We have done. (P6 L25)

P6 L32, “four models in ISMIP. Thus...” → four models in ISMIP, indicating that the biases... are almost the same.
We have done. (P7 L1)

P7 L3, “include... observation” → include members showing a spatial pattern of low similarity to that of observations.
We have done. (P7 L4)

P7 L7 “would be related” → could be related
Supplement 5 has been changed from the ratio to the absolute value. When the temperature bias is expressed by a ratio, the bias is relatively large in regions with a low temperature. The previous figure indicated such a tendency and it could give a misinterpretation. The sentence pointed out was removed with this modification and this does not impact on the other discussions.

P7 L16 “ has not much resembled the observation” → does not resemble that of the observations.

We have done. (P7 L16)

P7 L25 "change of " → changes in
  We have done. (P7 L24)

P7 L 26 "projected increments of the temperature". Ambiguous. I suggest → average rate of temperature increase per year, calculated from the 20-year period for each model
  We have modified to the below. (P7 L25)
"projected temperature increments, calculated from the average over the 20-year period for each model"

P7 L30, Any suggestion of why this result is different from other studies? It would be good to include some mention of this in the discussion section.
  First of all, the previous studies have discussed using the variables not temperature. To prevent this confusion, we have changed the sentence,
"the previous studies have showed the reduction of the uncertainty"
 → "the previous studies have showed the reduction of their projection uncertainties." (P7 L30)
Regarding the reason of the difference, the skill score for temperature is around 0.9 even at the maximum except Central America and the spatial pattern is quite similar among the models. Thus, the model selection using the score hardly has an impact on the reduction of uncertainty. On the other hand, the difference in the maximum bias between full ensemble and subset is not so small. In term of the quantity, we expect a weak relation between the performance and the future change. It is consistent with the suggestions by previous studies that the performance of the present climate simulations is not necessarily related to the uncertainties of future projections (e.g., Knutti, 2010, Shiogama et al., 2011). Thus, we have added the followings to P7 L31.
"This is because the spatial pattern for the temperature is quite similar among the models and then the model selection using the score hardly has an impact on the reduction of uncertainty. On the other hand, the difference in the bias between the full set and the subset is large. The previous studies have suggested that the performance of the present climate simulations is not necessarily related to the uncertainties of future projections (e.g., Knutti, 2010, Shiogama et al., 2011) and we expected such a relation between the quantitative performance and the future change in this study."
Shiogama H, S Emori, N Hanasaki, M Abe, Y Masutomi, K Takahashi, T Nozawa: Observational constraints indicate risk of drying in the Amazon basin. Nature Comm., 2, 253, doi: 10.1038/ncomms1252, 2011.

P8 L2 "with totally covering" → and cover the full range...
  We have done. (P8 L7)

P8 L5 "like at least the bias.." → for example whether we use the bias or the skill score.
  We have done. (P8 L10)

P8 L7, "capture the full range less... " → capture less than 60% of the full range in all regions. P8 L8, capture the wider → capture a wider range
  We have done. (P8 L12, P8 L13)

P8 L8, "subsets, differing from..." → subsets, a result markedly different than for ΔT, where both CORDEX and ISIMIP have relatively large coverage.
  We have done. (P8 L13)

P8 L9, shows the difficulty of → has difficulty capturing
  We have done. (P8 L15)

P8 L12, → uncertainty range (maximum-minimum) is ...
  We have done. (P8 L18)

P8 L16, Only in... → In Central Asia, the full range of.. remains below the 25th percentile... while the

maximum-minimum range of .. adequately covers the IQR... Thus, the three models.. Central Asia underestimate the average tendency...
We have done. (P8 L22)

P8 L19 "despite being... differing from ISIMIP" This is unclear. Please rephrase.
The sentence has rephrased to,
"despite that, differing from ISIMIP, CORDEX can select suitable models for discussion of climate change in Central Asia." (P8 L25)

P8 L24 "high coverage for the temperature.." How about precipitation? Please include the precipitation results for ISIMIP too.
Thank you for your suggestions. We have added the following sentence,
"and low coverage for the precipitation change in more than 60% of all regions" (P8 L31)

P8 L 31 presents distribution → represents the distribution...
We have done. (P9 L7)

P9 L32, "They focused... models" Unclear, please rephrase. In particular, "grid" is ambiguous, I suggest using model (or grid cell) if applicable.
We have rephrased to,
"They estimated the coverages in each number of models to investigate the change of coverage performance of the subset with most widely covering the uncertainty in each of the global grid cells or the regional cells." (P10 L6)

P10 L8, "which are less than nine members" this fragment does not add anything. Please remove or adjust to explain why nine members is relevant.
Thank you for pointing out. The sentence was removed.

P10 L24, "high performed models" → high performance models.
We have done. (P10 L31)

P10 L25, "regarding to" → with regard to
We have done. (P10 L32)

P10 L29, "13 models in 25 all high performance models" → 13 high performance model (out of 25) are included...
We have done. (P11 L2)

P10 L32 "Therefore, although... possibility. " Unclear, please rephrase.
We have rephrased the sentence,
"For one variable, there is a possibility of 50% that a model with a small bias shows a high skill score but it is difficult to select such models for both variables of temperature and precipitation." (P11 L6)

P11 L3, the subset → their model ensemble indicates...
We have done. (P11 L9)

P11 L7, the dataset with reasonable for... projections → a dataset with reasonable values for... and with enough coverage of the projection uncertainties.
We have done. (P11 P12)

P11 L8 with based → with values based
We have done. (P11 L14)

P11 L12, Remove "in the case". Also, can you suggest other relevant variables, given that you state temperature and precipitation are not enough?

We have added "the pattern of circulation, indicated by sea level pressure (SLP) and geopotential height" to P11 L25.

P11 L19, similar to the above, can you include a suggestion of a variable to characterize "circulation" (e.g. geopotential height, sea level pressure...)?
We have added "the pattern of circulation, indicated by sea level pressure (SLP) and geopotential height" to P11 L25.

P11 L23, similar to the above, simply stating "the other ones" is too vague. Make specific suggestions of potentially useful variables.
We have modified around this sentence according to general comment (i). Please check it.

P12 L3, How about using both precipitation and temperature? Can you comment here on a combined evaluation and whether the results are likely to be different from the ones included in this study?
As indicated in P11 L3-8, there is an extremely small number of models showing a high performance for both temperature and precipitation. We have added the following line to the end of the line pointed out. (P12 L15)
"A combined evaluation for both temperature and precipitation remains difficulty in obtaining an adequate number of models."

P12 L20, "with capturing" → while capturing
We have done. (P12 L33)

P12 L22, "the global common subset" → the global common (ISIMIP) subset ; "but large model ensemble is needed" → but performs better when a large number of models is used.
We have done. (P13 L2)

P12 L26, "need to be changed" to what? Specify, e.g. increased to seven, or nine models if possible....
As you mentioned, we have modified. (P13 L7)

P12 L31, specify which other variable might be relevant.
We have added the followings to P13 L12.
"(e.g. the circulation patterns shown by sea level pressure and geopotential height)"

Figure 4: The red dot is missing for Central Asia. Is it also an overlap? If so, change the caption accordingly. For the 3 regions where the dots overlap, I suggest changing the size of one of the dots (e.g. the red dot slightly larger than the blue dot), so that the outside of the red dot is visible, this would make the plot easier to understand by showing the overlap, rather than having to read the caption.
Thank you for your suggestions. We have modified the figure with changing the size of the dots.
* * *
We have revised out manuscript to address comments.

[revised manuscript text omitted]

† denotes the ISIMIP2b model.

[Figure]

**Supplement 2: Regional classification defined in CORDEX. (Coordinate information is available from: http://www.cordex.org/domains/; last accessed 8 Nov. 2018).**

[Figure]

Supplement 3: Models with the top 50% of the CMIP5 models for the model bias and Taylor's skill score in each CORDEX region. The numbers on the x-axis correspond to the individual model number in the bottom box. The y-axis denotes the models with low bias and with high score for temperature (T(CMIP$_{lowB}$) and T(CMIP$_{highS}$)) and for precipitation (P(CMIP$_{lowB}$) and P(CMIP$_{highS}$)) from the upper to the bottom. Gray square indicates the models fits the condition on the y-axis and black square indicates the inclusion in the CORDEX subset. Light gray bar in Antarctica indicates the missing data because of missing the CRU reference data.

[Figure]

**Supplement 4: Skill score for annual mean model precipitation over land. Reference data are from GPCC. The whiskers of the box plots show the range between the maximum and the minimum scores. The boxes and the lines within the boxes indicate the 25th to 75th percentile range and the median, respectively. Green plots indicate the spread of the score of six observed data; CRU, CPC, PRECL, CMAP, GPCP 1dd and MSWEP. The other plots indicate the model bias in the full set of 50 CMIP5 model set (black), the model sets with the top 50% of the CMIP5 models for the bias (yellow) or Taylor's skill score (orange) and the model sets selected for ISIMIP (blue) and CORDEX (red).**

[Figure]

[Figure]

[Figure]

5    **Supplement 5: Annual mean model temperature bias over land (K). Reference data are from CRU TS. The whiskers of the box plots show the range between the maximum and the minimum biases. The boxes and the lines within the boxes show the 25th to 75th percentile range and the median, respectively. The other plots indicate the model bias in the full set of 50 CMIP5 model set (black), the model set with the top 50% of the full set for the bias (yellow) or Taylor's skill score (orange), and the model sets selected for ISIMIP (blue) and CORDEX (red). There are no plots over Antarctica because of missing the CRU reference data.**

[Figure]

**Supplement 6: As for Supplement 4, but for the skill score for the annual mean model temperature over land. There are no plots over Antarctica because of missing the CRU reference data.**

---

## Author Response (AR3)

Response to the comments from Editor for the manuscript:
"Uncertainties in climate change projections covered by the ISIMIP and CORDEX model subsets from CMIP5" by Ito et al.

We appreciate your decision and additional careful review. We have revised our manuscript in accordance with your pointing out. Editor comments below are shown in black and our responses are in blue. Page and line numbers are corresponding to the number in the revised manuscript.

-- Please use 'Table' and 'Figure' identifiers in your supplement as follows:
Supplement 1 --> Table S1
Supplement 2 --> Figure S1
Supplement 3 --> Figure S2
Supplement 4 --> Figure S3
Supplement 5 --> Figure S4
Supplement 6 --> Figure S5

and use those in the main manuscript. E.g. p4. l26
'The number of GCMs used in each of the defined regions is listed in Table 1, and each GCM is listed in Table S1.'

Thank you for your careful review. We have modified the identifier throughout our manuscript.
(P4 L26, P4 L28, P6 L12, P6 L25, P7 L4, P7 L9, P7 L14, P11 L3 and supplementary)

-- Please remove the supplement section from your main manuscript (p14.l3-l9)

We have removed the sentences. (P14 L3)

You may also consider adding acknowledgements to two anonymous reviewers for their help on improving the manuscript to your Acknowledgements section.

Thank you for your suggestions. We have added the acknowledgements to two reviewers.

In this revision, a few words without your requests have been corrected simply. Please check the following. P7 L28, P8 L3, P11 L29, P11 L31, P12 L1
* * *

[revised manuscript text omitted]

**Table S1: List of CMIP5 GCMs used in CORDEX.**

| Model / Region | North America | Central America | South America | Europe | MED | MENA | Africa | Central Asia | East Asia | South Asia | SEA | Australasia | Arctic | Antarctica |
|---|---|---|---|---|---|---|---|---|---|---|---|---|---|---|
| ACCESS1.0 | | | | | | | | | | ○ | ○ | ○ | | ○ |
| ACCESS1.3 | | | | ○ | | | | | | | | ○ | | |
| BNU-ESM | | | | | | | ○ | | | | | | | |
| CCSM4 | | | | ○ | | | | | | ○ | ○ | ○ | | ○ |
| CMCC-CM | | | | | ○ | | | | | | | | | |
| CNRM-CM5 | | ○ | | ○ | ○ | ○ | ○ | | ○ | ○ | ○ | ○ | | ○ |
| CSIRO-Mk3.6.0 | | ○ | ○ | ○ | | | | | | ○ | ○ | ○ | | |
| CanESM2 | ○ | ○ | ○ | ○ | | | ○ | | | ○ | ○ | ○ | ○ | |
| EC-EARTH | ○ | ○ | ○ | ○ | | ○ | ○ | | ○ | ○ | ○ | ○ | ○ | ○ |
| FGOALS-g2 | | | | | | | | | ○ | | | | | |
| FGOALS-s2 | | | | | | | ○ | | | | | | | |
| GFDL-CM3 | | | | | | | | | | ○ | | ○ | | ○ |
| GFDL-ESM2G | | | | | | | ○ | | | | | | | |
| GFDL-ESM2M† | ○ | ○ | ○ | ○ | | ○ | ○ | ○ | | ○ | | | | |
| GISS-E2-R | | | | ○ | | | | | | | | | | |
| HadGEM2-AO | | | | | | | | | ○ | | | ○ | | |
| HadGEM2-CC | | | | | | | | | | | | ○ | | |
| HadGEM2-ES† | ○ | ○ | ○ | ○ | ○ | ○ | ○ | ○ | ○ | ○ | ○ | ○ | | ○ |
| IPSL-CM5A-LR† | | | | | | | | | | ○ | ○ | | | |
| IPSL-CM5A-MR | | ○ | ○ | ○ | ○ | | ○ | | | ○ | | | | |
| MIROC-ESM | | | | | | | ○ | | | | | | | |
| MIROC-ESM-CHEM | | | | | | | ○ | | | | | | | |
| MIROC5† | | ○ | ○ | ○ | | | ○ | | | ○ | | ○ | | |
| MPI-ESM-LR | ○ | ○ | ○ | ○ | ○ | ○ | ○ | | ○ | ○ | ○ | ○ | ○ | ○ |
| MPI-ESM-MR | ○ | | | | | | | | ○ | ○ | ○ | | ○ | |
| MRI-CGCM3 | | | | | | | ○ | | | | | | | ○ |
| MRI-AGCM60 | | | | | | | | | ○ | | | ○ | | |
| NorESM1-M | | ○ | ○ | ○ | | | ○ | | | ○ | | ○ | ○ | ○ |

† denotes the ISIMIP2b model.

[Figure]

5    **Figure S1: Regional classification defined in CORDEX. (Coordinate information is available from: http://www.cordex.org/domains/; last accessed 8 Nov. 2018).**

[Figure]

| 1 ACCESS1-0 | 11 CMCC-CESM | 21 FGOALS-g2 | 31 GISS-E2-R-CC | 41 MIROC-ESM-CHEM |
|---|---|---|---|---|
| 2 ACCESS1-3 | 12 CMCC-CM | 22 FGOALS-s2 | 32 HadCM3 | 42 MIROC4h |
| 3 bcc-csm1-1 | 13 CMCC-CMS | 23 FIO-ESM | 33 HadGEM2-AO | 43 MIROC5 |
| 4 bcc-csm1-1-m | 14 CNRM-CM5 | 24 GFDL-CM2p1 | 34 HadGEM2-CC | 44 MPI-ESM-LR |
| 5 BNU-ESM | 15 CNRM-CM5-2 | 25 GFDL-CM3 | 35 HadGEM2-ES | 45 MPI-ESM-MR |
| 6 CCSM4 | 16 CSIRO-Mk3-6-0 | 26 GFDL-ESM2G | 36 inmcm4 | 46 MPI-ESM-P |
| 7 CESM1-BGC | 17 CSIRO-Mk3L-1-2 | 27 GFDL-ESM2M | 37 IPSL-CM5A-LR | 47 MRI-CGCM3 |
| 8 CESM1-CAM5 | 18 CanCM4 | 28 GISS-E2-H | 38 IPSL-CM5A-MR | 48 MRI-ESM1 |
| 9 CESM1-FASTCHEM | 19 CanESM2 | 29 GISS-E2-H-CC | 39 IPSL-CM5B-LR | 49 NorESM1-M |
| 10 CESM1-WACCM | 20 EC-EARTH | 30 GISS-E2-R | 40 MIROC-ESM | 50 NorESM1-ME |

**Figure S2: Models with the top 50% of the CMIP5 models for the model bias and Taylor's skill score in each CORDEX region. The numbers on the x-axis correspond to the individual model number in the bottom box. The y-axis denotes the models with low bias and with high score for temperature (T(CMIP$_{lowB}$) and T(CMIP$_{highS}$)) and for precipitation (P(CMIP$_{lowB}$) and P(CMIP$_{highS}$)) from the upper to the bottom. Gray square indicates the models fit the condition on the y-axis and black square indicates the inclusion in the CORDEX subset. Light gray bar in Antarctica indicates the missing data because of missing the CRU reference data.**

[Figure]

| 5 | **Figure S3: Skill score for annual mean model precipitation over land. Reference data are from GPCC. The whiskers of the box plots show the range between the maximum and the minimum scores. The boxes and the lines within the boxes indicate the 25th to 75th percentile range and the median, respectively. Green plots indicate the spread of the scores of six observed data: CRU, CPC, PRECL, CMAP, GPCP and MSWEP. The other plots indicate the score in the full set of 50 CMIP5 model set (black), the model sets with the top 50% of the CMIP5 models for the model bias (yellow) or the score (orange) and the model sets selected for ISIMIP (blue) and CORDEX (red).** |

[Figure]

5    **Figure S4: Annual mean model temperature bias over land (K). Reference data are from CRU. The whiskers of the box plots show the range between the maximum and the minimum biases. The boxes and the lines within the boxes show the 25th to 75th percentile range and the median, respectively. The other plots indicate the model bias in the full set of 50 CMIP5 model set (black), the model set with the top 50% of the full set for the bias (yellow) or Taylor's skill score (orange), and the model sets selected for ISIMIP (blue) and CORDEX (red). There are no plots over Antarctica because of missing the CRU reference data.**

[Figure]

5   **Figure S5: As for Figure S3, but for the annual mean model temperature over land.  There are no plots over Antarctica because of missing the CRU reference data.**